# Anomalous thickness dependence of Curie temperature in air-stable two-dimensional ferromagnetic 1T-CrTe₂ grown by chemical vapor deposition

Lingjia Meng[1,2,13], Zhang Zhou [3,13], Mingquan Xu[4,5,13], Shiqi Yang[6,7,13], Kunpeng Si[1], Lixuan Liu[1], Xingguo Wang[1], Huaning Jiang[1], Bixuan Li[1,2], Peixin Qin[1], Peng Zhang[1], Jinliang Wang[2], Zhiqi Liu [1], Peizhe Tang [1,8], Yu Ye [6,9✉], Wu Zhou [4,5,10✉], Lihong Bao [3,10,11✉], Hong-Jun Gao [3,10,11] & Yongji Gong [1,12✉]

The discovery of ferromagnetic two-dimensional van der Waals materials has opened up opportunities to explore intriguing physics and to develop innovative spintronic devices. However, controllable synthesis of these 2D ferromagnets and enhancing their stability under ambient conditions remain challenging. Here, we report chemical vapor deposition growth of air-stable 2D metallic 1T-CrTe₂ ultrathin crystals with controlled thickness. Their long-range ferromagnetic ordering is confirmed by a robust anomalous Hall effect, which has seldom been observed in other layered 2D materials grown by chemical vapor deposition. With reducing the thickness of 1T-CrTe₂ from tens of nanometers to several nanometers, the easy axis changes from in-plane to out-of-plane. Monotonic increase of Curie temperature with the thickness decreasing from ~130.0 to ~7.6 nm is observed. Theoretical calculations indicate that the weakening of the Coulomb screening in the two-dimensional limit plays a crucial role in the change of magnetic properties.

[1] School of Materials Science and Engineering, Beihang University, 100191 Beijing, P. R. China. [2] School of Physics, Beihang University, 100191 Beijing, P. R. China. [3] Institute of Physics and University of Chinese Academy of Sciences, Chinese Academy of Sciences, 100190 Beijing, P. R. China. [4] School of Physical Sciences, University of Chinese Academy of Sciences, 100190 Beijing, P. R. China. [5] CAS Key Laboratory of Vacuum Physics, University of Chinese Academy of Sciences, 100190 Beijing, P. R. China. [6] State Key Laboratory for Mesoscopic Physics and Frontiers Science Center for Nano-Optoelectronics, School of Physics, Peking University, 100871 Beijing, P. R. China. [7] Academy for Advanced Interdisciplinary Studies, Peking University, 100871 Beijing, P. R. China. [8] Center for Free-Electron Laser Science, Max Planck Institute for the Structure and Dynamics of Matter, Hamburg 22761, Germany. [9] Collaborative Innovation Center of Quantum Matter, 100871 Beijing, P. R. China. [10] CAS Center for Excellence in Topological Quantum Computation, University of Chinese Academy of Sciences, 100190 Beijing, P. R. China. [11] Songshan Lake Materials Laboratory, Dongguan 523808 Guangdong, P. R. China. [12] Center for Micro-Nano Innovation of Beihang University, 100191 Beijing, P. R. China. [13] These authors contributed equally: Lingjia Meng, Zhang Zhou, Mingquan Xu, Shiqi Yang. ✉email: ye_yu@pku.edu.cn; wuzhou@ucas.ac.cn; lhbao@iphy.ac.cn; yongjigong@buaa.edu.cn

Two-dimensional (2D) layered materials have attracted extensive research interests due to their marvelous electrical, mechanical, thermal, and optical properties[1–10]. In recent years, the discovery of 2D ferromagnets has also provided an ideal platform for exploring and understanding magnetism in low-dimensional systems[3,6–9,11–17], for example, the critical behavior and dimensional crossover of magnetic ordering[3,6,7]. To date, 2D magnetism has been observed primarily in mechanically exfoliated materials[5,7–9,18–20] grown by chemical vapor transport[6,21–26] or thin layers grown by molecular beam epitaxy (MBE)[14,15]. As exfoliation produces crystalline flakes randomly (the geometry and thickness cannot be well controlled) and MBE strictly requires lattice-matched substrates, these methods encounter high fabrication cost or low throughput. This becomes the largest barrier toward practical applications of these 2D ferromagnets. As a facile and industrially compatible method, chemical vapor deposition (CVD) has been widely used in the growth of 2D materials and the construction of their heterostructures[27–33]. Layered (such as $VTe_2$[31]) and non-layered (such as $FeTe$[30], $Cr_2S_3$[32], $CrSe$[33], and $\varepsilon$-$Fe_2O_3$[17]) magnetic crystals have been successfully prepared using typical CVD methods. However, due to the uncertainty of the growth mechanism of 2D layered magnetic materials and the limited characterization of their magnetic properties, the controllable synthesis of 2D magnetic materials via CVD remains a cutting-edge topic.

Besides the challenges in sample fabrication, the poor environmental stability of the reported 2D magnetic materials also restricts the exploration of their exotic properties, and potential industrial applications[6,7,30]. Many atomically thin 2D magnets confront rapid degradation when exposed to air[7], fatally limiting their potential applications. Previous reports have demonstrated that bulk 1T-$CrTe_2$ is a layered ferromagnetic (FM) compound[34] with a high Curie temperature ($T_c$) of about 310 K[35,36]. Facile synthesis of thickness-controlled 1T-$CrTe_2$ with good stability, which is encapsulation-free and operable in air, is thus highly desirable and yet to be developed in the field of 2D ferromagnets.

In this work, we developed a CVD strategy to synthesize 1T-$CrTe_2$ on $SiO_2$/Si substrates with controlled thickness by controlling the growth temperature and atmospheric condition. Robust anomalous Hall effect (AHE) is observed in the resulting samples without any encapsulation, indicating its FM properties and good stability. Furthermore, as the thickness of 1T-$CrTe_2$ reducing from tens of nanometers to several nanometers, the easy axis changes from in-plane to out-of-plane, and a monotonic increase of $T_c$ is observed. Theoretical calculations indicate that the Coulomb screening plays a crucial role in the change of magnetic properties.

## Results

**Controlled growth of 1T-$CrTe_2$ single crystals.** Herein, ultrathin 1T-$CrTe_2$ single crystals were directly grown on $SiO_2$/Si substrates at a temperature of approximately 983 K via CVD, using $CrCl_2$ and Te powders as the precursors (Fig. 1a). The thickness of the resulting sample is very sensitive to growth temperature and growth atmosphere. For instance, increasing the growth temperature from 973 to 993 K, the averaged thickness ($d$) of the single crystals increases from about 1.2 nm (bilayer) to about 47.9 nm (Fig. 1b and Supplementary Fig. 1). Furthermore, under the same growth temperature, increasing the hydrogen concentration within a certain range will greatly reduce the thickness (Supplementary Fig. 2). Through a rational combination of both the growth temperature and reaction atmosphere, we demonstrate the layer-controlled synthesis of 1T-$CrTe_2$ single crystals (Fig. 1b and Supplementary Fig. 3). The statistical studies demonstrate that the layer-controlled CVD synthesis of ultrathin 1T-$CrTe_2$

single crystal is relatively precise (Supplementary Fig. 3). More details on sample synthesis are described in the Methods section. Optical microscopy (OM) images show that the obtained 1T-$CrTe_2$ crystals usually present a hexagonal morphology (Fig. 1b and inset in Fig. 1c) with the lateral size ranging from a few to tens of micrometers. Atomic force microscopy (AFM) measurement on a representative stepped single crystal (Fig. 1c) showed the height to be ~1.2 nm, indicating a bilayer crystal. A step height of about 0.6 nm can be observed in some samples (Supplementary Fig. 4), confirming the layered nature of 1T-$CrTe_2$[35,36]. The as-grown sample with the empirical formula of $CrTe_2$ is confirmed by high-resolution X-ray photoelectron spectroscopy (XPS) (Supplementary Fig. 5). The two main characteristic vibration modes of $E_{2g}$ and $A_{1g}$, located at 123.6 and 143.6 $cm^{-1}$, respectively, are observed in the Raman spectra for the as-synthesized samples (Supplementary Fig. 6). In addition, we plot the spatial Raman intensity mapping of the 123.6 $cm^{-1}$ mode across the entire 1T-$CrTe_2$ crystal in the inset of Fig. 1b, which shows uniform color contrast, indicating that the crystal is highly homogeneous.

**Air-stability of 1T-$CrTe_2$ single crystals.** It is worth noting that these two Raman characteristic peaks of the few-layered 1T-$CrTe_2$ single crystal hold similar intensity even after the sample was exposed to the ambient environment for 5 days (Fig. 1d and Supplementary Fig. 7), indicating the preservation of high crystallinity for the obtained 1T-$CrTe_2$ samples against the ambient exposure. This air-stability of the as-obtained 1T-$CrTe_2$ ultrathin single crystals is significant for exploring applications of 2D magnets in future spintronic devices.

**Structural characterization of 1T-$CrTe_2$ single crystals.** Aberration-corrected scanning transmission electron microscopy (STEM) was performed to characterize the atomic structure of the synthesized 1T-$CrTe_2$ single crystals. The low-magnification STEM high-angle annular dark-field (HAADF) image (Fig. 2a) displays a typical hexagonal-shape 1T-$CrTe_2$ flake and the energy-dispersive X-ray spectroscopy (EDS) analysis (Supplementary Fig. 8) indicates the homogeneous distribution of Cr and Te atoms across the entire crystal, which is consistent with the Raman mapping (inset in Fig. 1b). The well-preserved morphology after wet transfer further confirms the stability of the sample. The spatial distribution of Cr and Te columns can be easily discriminated from the Z-contrast STEM-HAADF images, as the Te atom columns with higher atomic number (Z) present brighter contrast than Cr columns with a lower atomic number. The atomic-resolution STEM-HAADF image (Fig. 2b) clearly shows that each Cr atomic column is surrounded by six Te atom columns arranging into a hexagonal lattice, consistent with the in-plane atomic configuration of the 1T phase. The corresponding fast Fourier transformation pattern of Fig. 2b can be indexed as the [001] zone axis of 1T-$CrTe_2$. The cross-sectional STEM-HAADF imaging and EELS mapping (Fig. 2d–f and Supplementary Fig. 8d–g) reveal that each monolayer slab consists of one layer of Cr atoms sandwiched between two layers of Te atoms, arranging into a characteristic Z-shaped construction, which matches well with the nature of octahedral coordination in the 1T phase. The schematic of the 1T-$CrTe_2$ crystal structure is shown in the insets of Fig. 2b, d with Cr in maroon and Te in yellow. The atomic-scale STEM analysis proves that the as-grown crystals are 1T-$CrTe_2$ with the AA stacking order. No obvious intercalated atoms were detected between two individual layers. From the in-plane STEM Z-contrast image and cross-sectional HAADF image analysis, we conclude that the as-synthesized 1T-

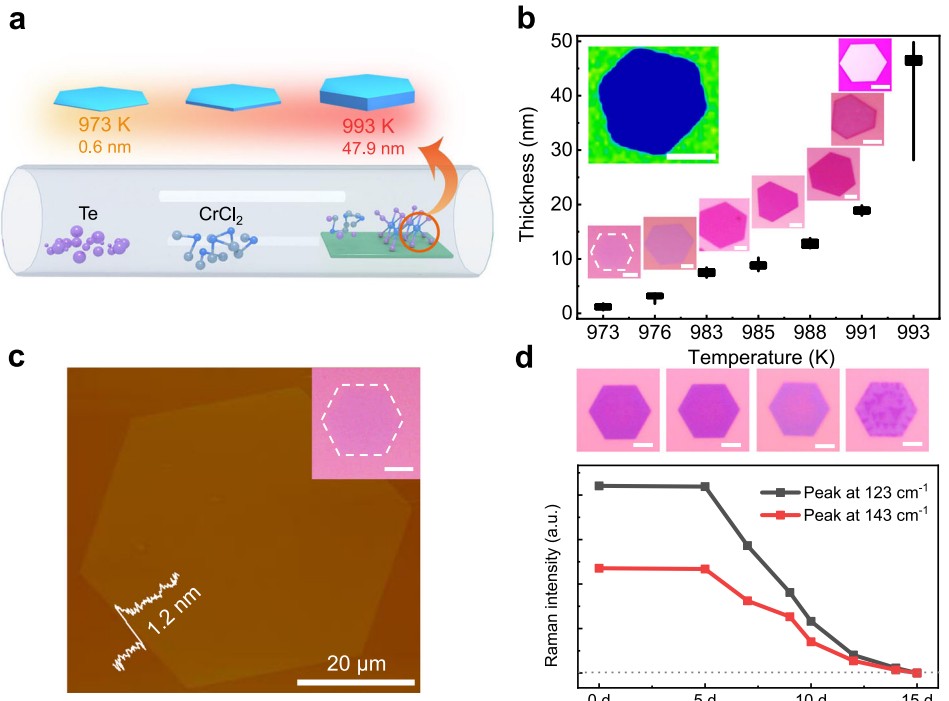

**Fig. 1 Synthesis and characterization of 1T-CrTe₂ single crystals on SiO₂/Si substrates. a** Schematic of the grown 1T-CrTe₂ single crystals by the CVD method. $CrCl_2$ and Te powders are used as Cr and Te sources, respectively, to realize the reaction at a relatively low temperature. **b** The sample thickness as a function of growth temperature with the corresponding OM images. The black line and the black rectangle indicate the range of the thickness and the averaged thickness of the samples grown at a given temperature, respectively. The top-left inset shows the Raman mapping image of 1T-CrTe₂ based on the vibration mode at 123.6 cm$^{-1}$. The thickness of the sample is about 10.0 nm. **c** AFM image and the corresponding OM image (upper right inset) of a typical 1T-CrTe₂ hexagonal nanoflake on a 285 nm SiO₂/Si substrate. The height profile in the lower left inset shows that this sample flake is ~1.2 nm thick. **d** Environmental stability investigation based on a ~4-nm thick sample. Upper panels are OM images of 1T-CrTe₂ samples that are exposed in the ambient environment for 0, 5, 10, and 15 days, respectively. Below is the evolution of Raman intensity with time progression. The gray dotted line indicates the zero Raman signal. Scale bars in **b**, **c**, and **d**: 20 μm.

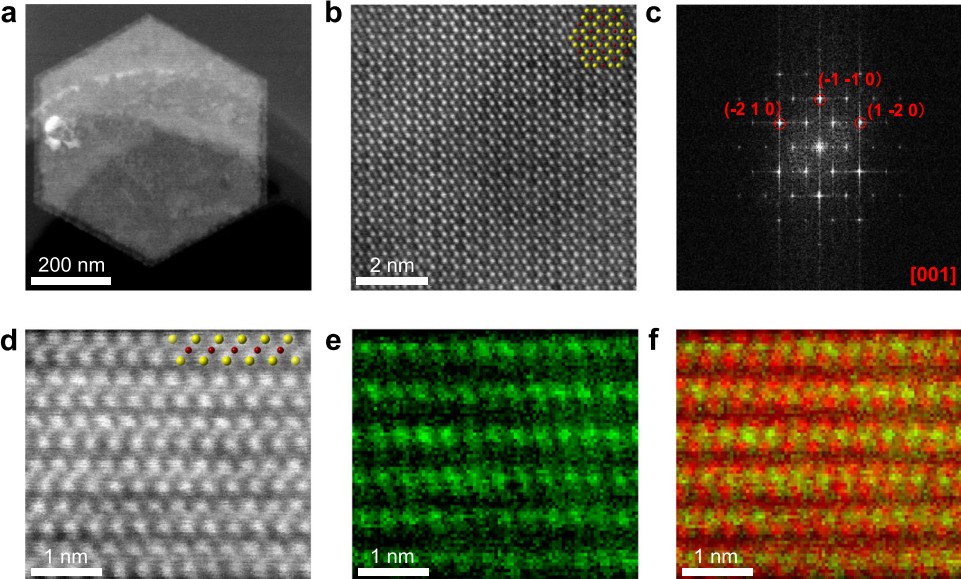

**Fig. 2 STEM analysis of the hexagonal 1T-CrTe₂ sample. a** Low-magnification STEM-HAADF image. **b** Atomic-resolution STEM-HAADF image and **c** the corresponding FFT pattern. **d** Atomic-resolution STEM-HAADF image of the cross-sectional specimen, showing an obvious layered structure. Insets in **b**, **d**: the corresponding crystal structures, with Cr in maroon and Te in yellow. **e** STEM-EELS elemental mapping of Cr. **f** The overlaid image of Cr-EELS map (in green) on HAADF image (in red).

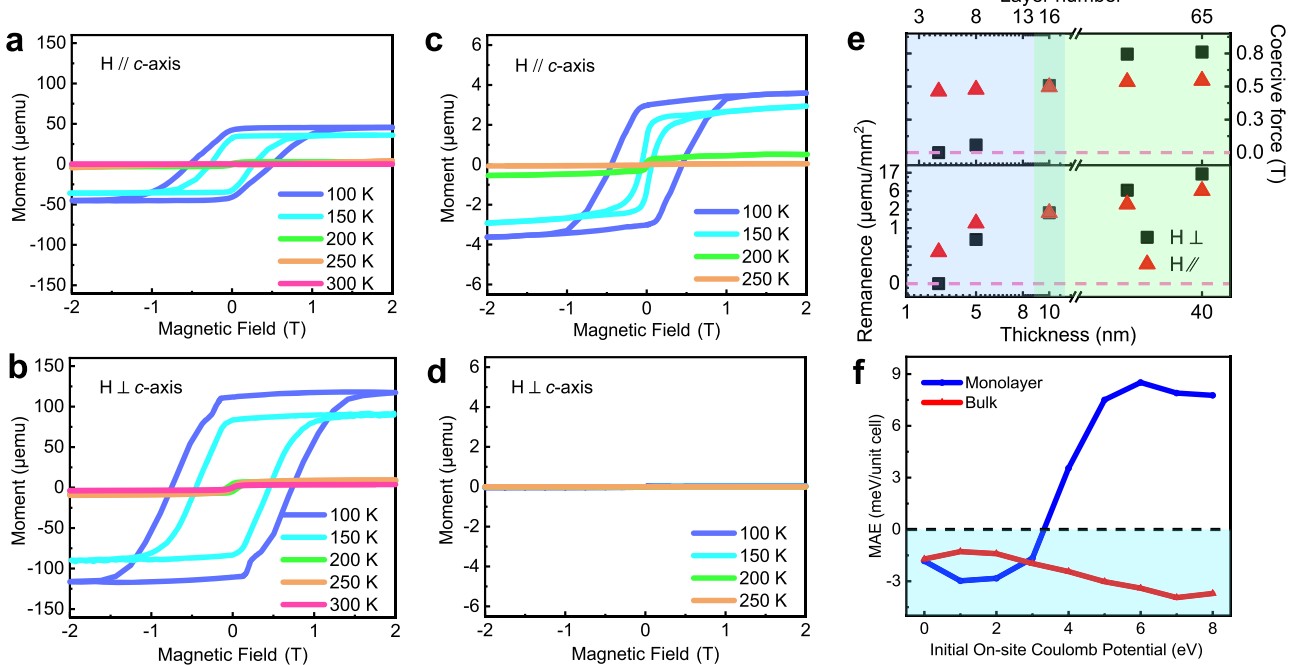

**Fig. 3 Thickness-dependent magnetic anisotropy of 1T-CrTe₂ crystals grown on SiO₂/Si substrate. a–d** Magnetic hysteresis loops for 1T-CrTe₂ flakes with a thickness of ~40.0 nm (**a**, **b**) and ~3.0 nm (**c**, **d**) under the magnetic field parallel (**a**, **c**) and vertical (**b**, **d**) to the *c*-axis of the crystal, respectively. The corresponding topographies and optical contrast images are displayed in Supplementary Fig. 9. **e** Remanence and coercive force at zero field for 1T-CrTe₂ samples of various thicknesses. The magnetic hysteresis loops at 100 K was used to extract the value of remanence and coercive force. The remanence is normalized by the sample area. **f** DFT-calculated magnetic anisotropy energy (MAE) as a function of initial on-site Coulomb potential (*U*) for monolayer and bulk 1T-CrTe₂, respectively. The MAE ($E_{MAE}$) is defined as $E_{MAE} = E_{out} - E_{in}$, in which $E_{out}$ and $E_{in}$ are the calculated total energies per unit cell with easy axis along with the directions of out-of-plane and in-plane, respectively.

CrTe₂ belongs to the space group $P\bar{3}m1$ and the lattice parameters are $a = b = 3.77$ Å, $c = 6.01$ Å.

**Thickness-dependent magnetic anisotropy of 1T-CrTe₂ crystals.** The thickness-dependent magnetic properties of the as-synthesized 1T-CrTe₂ samples were firstly examined through a vibrating sample magnetometer (VSM) and superconducting quantum interference device (SQUID) measurement under magnetic fields parallel and vertical to the *c*-axis of the crystal (Fig. 3a–d and Supplementary Fig. 9). In the thick sample (*d* ~ 40.0 nm), one can observe that the magnetization hysteresis loops (*M–H* curve) regardless of the applied magnetic field are along with parallel (Fig. 3a) or vertical (Fig. 3b) directions to the *c*-axis, indicating the long-range FM order in the as-grown samples. Notably, the sample (*d* ~ 40.0 nm) still has a small amount of remanence even at 300 K under the in-plane magnetic field (*H* ⊥ *c*-axis) (Fig. 3b), but shows no remanence above 200 K under the out-of-plane magnetic field (*H* // *c*-axis), indicative of an in-plane easy axis for thick 1T-CrTe₂ crystals. Similarly, the characteristic FM *M–H* loops at 10 K of CrTe₂ flakes with various thicknesses under in-plane magnetic field are observed[36]. Meanwhile, the Faraday measurement indicates that the *T*c in few-layered CrTe₂ is around 305 K[36], which is similar to the results from our SQUID measurement. Especially, a minor step-like feature at the low field is presented in the *M–H* loop (Fig. 3b), which is usually observed in complex magnetic systems with composite ordering. Noting that no obvious defects or stacking faults were observed from the STEM results and no step-like feature was founded in the thin samples. In this work, we infer that the additional minor step-like feature at low field may come from a complex magnetic domain structure[37]. In the thin sample (*d* ~ 3.0 nm), a prominent hysteresis

loop is observed under the magnetic field parallel to the *c*-axis (Fig. 3c), while nearly no remanence under in-plane magnetic field (Fig. 3d), showing that the easy axis in the ultrathin 1T-CrTe₂ crystal is along the out-of-plane direction. The slope changes at ~0 T under a relatively lower temperature may be caused by the weak magnetic signal from the ultrathin samples[30].

Figure 3e summarizes the magnetic properties of the 1T-CrTe₂ crystals as a function of thickness. With decreasing of the thickness *d* (from ~40.0 to ~3.0 nm), the magnetic easy axis changes direction from in-plane to out-of-plane, and the critical thickness is approximately 10.0 nm (Supplementary Fig. 10). The electrostatic coupling with the substrate might play a critical role in this phenomenon. To understand this behavior, we calculate the magnetic anisotropic energy (MAE) for bulk and monolayer 1T-CrTe₂ by changing the on-site Coulomb potential (*U*) on *d* orbitals of Cr atoms that are deeply influenced by screening effect. As shown in Fig. 3f and Supplementary Fig. 13, we observe the large MAE in the 2D limit that could resist the thermal fluctuation and host the long-range FM order in a monolayer sample. As *U* increases, MAE changes sign for monolayer 1T-CrTe₂; however, it remains negative for bulk sample. The value of MAE is determined by the spin-orbit coupling between *d* orbitals around the Fermi level. When we decrease the thickness of 1T-CrTe₂ to make the samples from three-dimensional (3D) to 2D, quantum confinement changes the electronic structures and varies the density of states (DOS) around the Fermi level (Fig. 4h), so MAE could change sign when *U* is larger than 4 eV. A similar phenomenon was observed in the Co thin film[38]. On the other hand, the distribution of *d* orbitals around the Fermi level also depends on its on-site *U* on each Cr atom. Besides, the effective Coulomb screening could be influenced by the dimension of the sample. Compared with that in 3D bulk, the

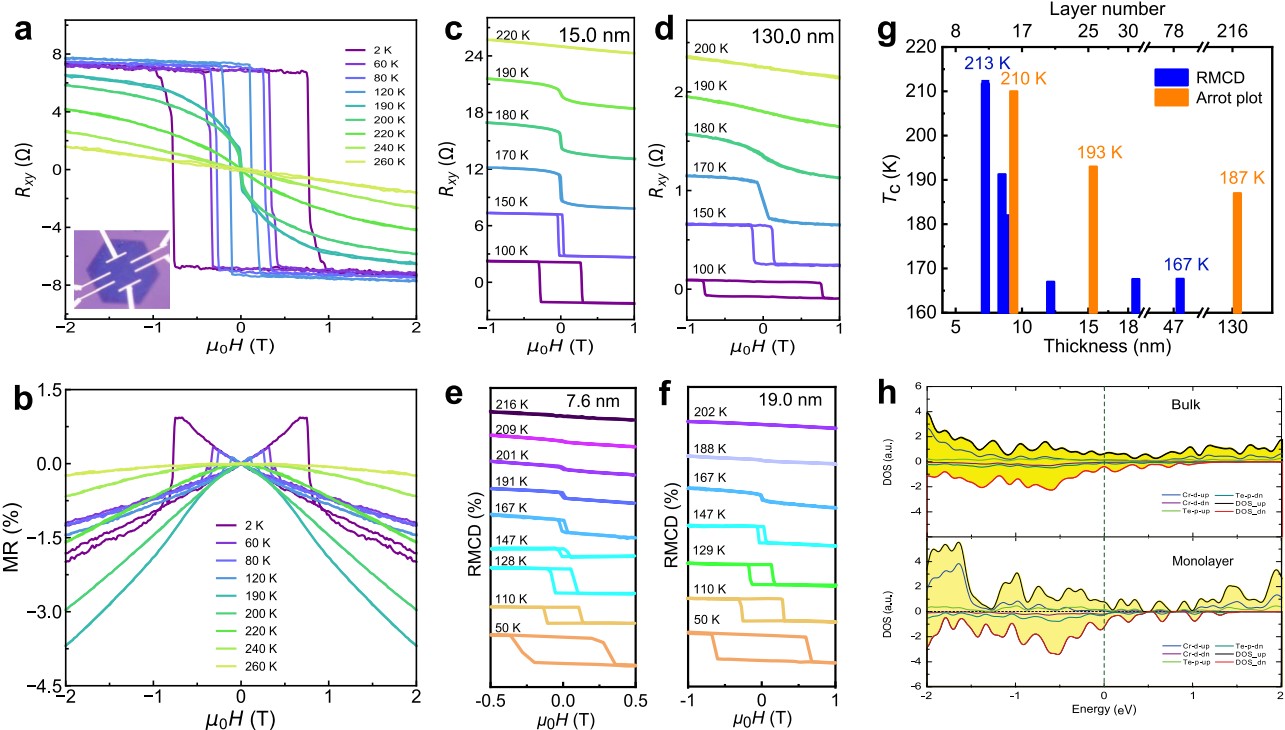

**Fig. 4 Magneto-transport and RMCD measurements of 1T-CrTe$_2$ single crystals. a**, **b** Out-of-plane Hall resistance hysteresis loops and magneto-resistance measured on the 10.0 nm thick 1T-CrTe$_2$ device. Inset: OM image of the corresponding Hall device. **c**, **d** Hall resistance at various temperatures obtained in 1T-CrTe$_2$ single crystals with a thickness of 15.0 and 130.0 nm, respectively. The corresponding MR are shown in Supplementary Fig. 15. **e**, **f** RMCD signal as a function of the out-of-plane magnetic field at different temperatures obtained in 1T-CrTe$_2$ domains with a thickness of 7.6 and 19.0 nm, respectively. **g** The trend of $T_c$ as a function of the layer number (and thickness) of 1T-CrTe$_2$. To minimize the effect of domain rotation and magnetic anisotropy, $T_c$ is determined from the Arrott plot (Supplementary Fig. 16). For RMCD, $T_c$ was extracted from the plots of temperature-dependent remanence of RMCD signal at zero field (Supplementary Fig. 18)[40]. $T_c$ values determined from RMCD and Arrott plots are presented in blue and orange, respectively. **h** The DFT-calculated density of states (DOS) for monolayer and bulk 1T-CrTe$_2$, respectively. The on-site Coulomb potential $U$ is 4 eV. The calculated Fermi levels are set to zero and highlighted by the green dotted line.

Coulomb screening is weakened in the atomic 2D thin film, resulting in a large $U$ from the electrostatic interaction with the substrate, which could eventually flip the easy axis.

**Thickness-dependent $T_c$ of 1T-CrTe$_2$ single crystals.** In addition to the magnetic anisotropy, we further estimate the $T_c$ for 1T-CrTe$_2$ with different thicknesses based on the SQUID measurement results (Supplementary Figs. 11 and 12). The $T_c$ was extracted from the inflection point defined by the minimum of $dM/dT$ in the $M$–$T$ curve[37]. The increase of $T_c$ from 179 to 189 K under an out-of-plane magnetic field is observed with reduced thickness from ~40.0 to ~3.0 nm, as further confirmed by magneto-transport and RMCD measurements, which is considered as a non-destructive and effective technique for probing the 2D magnetism[2,8,9]. The temperature-dependent longitudinal resistance curve ($R_{xy}$ – $T$) shows decreased resistance with temperature going down and a change of slope at ~200 K for a ~10-nm thick sample, indicating an FM metal (Supplementary Fig. 14). A clear anomalous Hall (AH) signal kink begins to appear at ~200 K, indicating a magnetic transition from paramagnetism to ferromagnetism (Fig. 4a). The robust AHE is observed in 1T-CrTe$_2$ van der Waals (vdW) FM single crystals, which has seldom been observed in other layered 2D materials grown by CVD. The near-vertical jumps of the AH signal versus perpendicular magnetic field below $T_c$ indicate a spin-flip transition. The coercive field, reaching ~1 T at 2 K, indicates hard ferromagnetism. The coercive field decreases with increasing

temperature, while the zero-field Hall resistance is almost the same over a wide temperature range (2–170 K), indicating the robust ferromagnetism in this temperature range. The calculated carrier mobility ($\mu$) and carrier concentration ($n$) based on the transport measurement are 9.1 cm$^2$V$^{-1}$s$^{-1}$ and 3.0 × 10$^{27}$ m$^{-3}$, respectively, further confirming a metallic state. Correspondingly, magneto-resistance (MR) exhibits a butterfly-shaped hysteresis phenomenon, with jumps occurring at the spin-flip transition magnetic fields (Fig. 4b and Supplementary Fig. 15). The negative MR is associated with the strong alignment of spins at high magnetic field, which suppresses the spin disorder and spin-dependent carrier scattering[39].

The occurrence of non-zero $R_{xy}$ at zero external magnetic field indicates the emergence of spontaneous magnetization, and the onset temperature marks the $T_c$ (Fig. 4c, d)[9,40,41]. The transition temperature from paramagnetism to ferromagnetism of thin flake ($d$ ~ 15.0 nm thick with $T_c$ ~ 193 K) is higher than that of the thick crystal ($d$ ~ 130.0 nm with $T_c$ of ~187 K) (Fig. 4c, d and Supplementary Fig. 16). Especially, the $T_c$ of the sample with thickness of ~10.0 nm is up to ~210 K, which further confirms an enhancement of $T_c$. The polar RMCD also verifies the FM order of the 1T-CrTe$_2$ ultrathin crystals (Supplementary Fig. 17). In addition, in the 7.5 nm thick crystal, the magnetic signal disappears at around 212 K (Fig. 4e), which is a significant increase compared to the thicker crystal ($d$ ~ 19.0 nm, $T_c$ ~ 167 K) (Fig. 4f). Both magneto-transport and RMCD measurements confirm the monotonic increase in $T_c$ with decreasing thickness ranging from ~130.0 to ~7.5 nm (Fig. 4g and Supplementary Fig. 18).

## Discussion

In order to confirm that the magnetic properties of 1T-CrTe$_2$ are intrinsic, we further investigated the magnetism of our samples with and without hexagonal boron nitride (h-BN) encapsulation by RMCD, respectively (Supplementary Fig. 19). To prevent the sample from air exposure, the samples were directly synthesized through a CVD equipment inside a glove box. The as-synthesized flakes show uniform color contrast under OM, indicating the uniform thickness of the 1T-CrTe$_2$ samples (Supplementary Fig. 19a). For these two samples, the nearly vertical jumps of the RMCD signals versus perpendicular magnetic field below $T_c$, along with the hysteresis loops shrinking as the temperature going up and vanishing at the magnetic transition point, indicate that the FM performance of the two 1T-CrTe$_2$ flakes is similar (Supplementary Fig. 19b, c). In addition, the extracted $T_c$ for the samples with and without h-BN encapsulation is ~184 and 185 K, respectively, further confirming the intrinsic ferromagnetism of the encapsulation-free samples. Moreover, we observe strong out-of-plane anisotropy with quite a large remanent magnetization along the z-direction in both the h-BN encapsulated and encapsulation-free sample, which is consistent with our previous observations. Thus, the experimental magnetic results from the unprotected 1T-CrTe$_2$ flakes are contributed by the intrinsic properties of this material.

Note that the extracted $T_c$ from AHE is consistently higher than the results from RMCD (Fig. 4g), which may result from many reasons, such as the laser heating during RMCD measurement, the existence of domains wall[9], and the doping effect during the nanofabrication[2]. Considering the complexity of the reasons for the discrepancy in $T_c$ extracted from AHE and RMCD, further efforts are needed to unravel the fundamental issue. Meanwhile, we do not observe magnetic signals from transport measurement in samples thinner than ~10.0 nm, which might be due to possible degradation occurring during nanofabrication. Nevertheless, the chemically assembled large-scale 2D ferromagnets with higher $T_c$ in thinner crystals (as determined by three different characterization methods) are interesting, but further detailed study is needed. This anomalous thickness-dependent $T_c$ is believed to be correlated with the layer-dependent magnetic anisotropy: the Coulomb screening effect in thin films is possible to changes the easy axis from in-plane to out-of-plane with decreased thickness, resulting in a higher $T_c$ at the out-of-plane direction and lower $T_c$ at the in-plane direction in atomically thin samples.

DOS near the Fermi level for bulk and monolayer 1T-CrTe$_2$ (Fig. 4h) is calculated to elucidate the strong dimensionality effect, confirming that 1T-CrTe$_2$ is an FM metal and revealing that the d orbitals of Cr atoms are partly occupied in all samples with different thicknesses. Such results imply that 1T-CrTe$_2$ is possibly an itinerant FM metal[36] even when the quantum confinement is considered. On the other hand, when we artificially change on-site $U$ that is determined by Coulomb screening, the occupations of Cr d orbitals change significantly especially in the 2D limit, resulting in the variation of DOS at the Fermi level. This effect could make 1T-CrTe$_2$ unsatisfied with the Stoner criteria, hinting at the possibility of other FM mechanisms. In future studies, some advanced numerical methods, such as the dynamical mean-field theory with constrained random phase approximation and density matrix embedding theory, are needed to be applied to this system, which could correctly describe the Coulomb screening in the 2D metallic system and may provide insights into the enhancement of $T_c$ with decreasing thickness in the special thickness range.

In summary, we have developed a CVD strategy to synthesize FM vdW 1T-CrTe$_2$ single crystals with various thickness. The processability in air and the observed robust AHE without any encapsulation for the samples with a thickness greater than 5 nm prove the relatively good stability of this material and keep the promise for its applications in future spintronic devices. 1T-CrTe$_2$, exhibiting layer-dependent magnetic anisotropy and anomalous thickness-dependent $T_c$, provides an ideal platform for the investigation of intriguing physical phenomena and offers the possibilities to fabricate high-temperature magnetoelectric devices in the 2D limit. Moreover, the simplicity and scalability of this CVD method provides an efficient strategy to construct 2D materials and 2D magnetic heterostructures with exciting magnetic properties, enabling opportunities for exploring both fundamental spin-related physical mechanisms and pioneering applications in spintronic devices.

## Methods

**Synthesis of 1T-CrTe$_2$.** 2D 1T-CrTe$_2$ single-crystal nanoflakes were synthesized on SiO$_2$/Si substrates (285-nm thick SiO$_2$, CETC) in a CVD system equipped with 2-inch quartz tube under atmospheric pressure. Divalent Chromous chloride (CrCl$_2$, 97%, Alfa Aesar) and tellurium (Te, 99.8%, Aldrich Chemistry) powders were used as Cr and Te sources, respectively. Specifically, 1 mg CrCl$_2$ powder was placed in a quartz boat that was put in the center of the heating zone of the furnace with a SiO$_2$/Si substrate placed face-down on the precursor. An alumina boat with 0.1 g Te powder was placed upstream. It is worth noting that the CrCl$_2$ is prone to hydrolysis in ambient conditions, so short exposure is required when loading the precursor into the furnace. After a ten-minute purge with ultrahigh purity argon gas (Ar, 99.999%), the furnace was first ramped to 983 K in 14 min and held at this temperature for another 2 min for sample growth. Te powder was placed at the edge of the tube where the temperature is about 703 K. Mixed gases of 1 sccm H$_2$ and 200 sccm Ar were used to provide a suitable atmosphere for the growth process. After the reaction completes, the furnace was quickly cooled down to 873 K and then naturally cooled down to room temperature in the Ar atmosphere.

**Sample characterization.** The morphology and thickness of the as-grown 1T-CrTe$_2$ single-crystal nanoflakes were characterized by an OM (CX44, Olympus) and an AFM (ICON, Veeco/Bruker) under the atmospheric environment. Raman spectra along with the corresponding mapping (DXRxi, Thermo) were performed under 532 nm laser excitation at room temperature. XPS (250Xi, Thermo Scientific Escalab) was used to characterize the chemical composition and chemical states of the samples. Temperature-dependent VSM measurement was performed in a Quantum Design Physical Property Measurement System (PPMS; Versalab system) with vibrational sample magnetometer utility and the high magnetic fields of up to 3 T. To avoid the possible degradation of the samples, polymethyl methacrylate (PMMA) coating was used before the VSM/SQUID test. The h-BN-encapsulated 1T-CrTe$_2$ samples were fabricated using the dry-transfer methods in a glove box[20]. PDMS was used as the transfer stamp without any heating during the deposition of thin flakes.

**Sample transfer.** To investigate the crystal structure at an atomic scale, the as-grown 1T-CrTe$_2$ was transferred onto copper grids for STEM characterizations. The transfer process is as follows: (1) home-made PMMA solution was spin-coated onto the sample at 4000 rpm for 60 s, followed by drying at 110 °C for 60 s; (2) SiO$_2$ layer was etched with 10% HF solution; (3) the PMMA layer was lifted off and rinsed with deionized water two times. It was then transferred to copper grids and air-dried; (4) acetone and isopropanol were used to remove the PMMA.

**STEM Z-contrast imaging and chemical analysis.** STEM characterization was performed on an aberration-corrected Nion HERMES-100 under the accelerating voltage of 100 kV, with a probe-forming angle of 30 mrad. The collection angle of STEM-HAADF imaging is 92–210 mrad. The EELS mappings were acquired under the same condition with a collection angle of 92 mrad and a probe current of ~60 pA. The cross-sectional specimens were prepared by focused ion beam using a standard lift-out procedure. The TEM-EDS analysis was performed on JEM-2100F, operating at 200 kV and equipped with an EDS system.

**RMCD analysis.** The polar RMCD measurements were performed based on the Attocube closed-cycle cryostat (attoDRY2100) down to 1.65 K and up to 9 T in the out-of-plane direction. The sample was moved by an x–y–z piezo stage (Piezo Positioning Electronic ANC300). A HeNe laser at 633 nm was used to generate linearly polarized light and was coupled into the system using free-space optics. The linearly polarized light was modulated between left and right circular polarization by a photoelastic modulator at 50.052 kHz. By using a high numerical aperture (0.82) objective, a Gaussian beam with a spot size of ~2 μm in diameter was focused onto the sample surface. The reflected light was also collected by the free-space optics and detected by a photomultiplier tube. Due to the polar magneto-optic effect, the magnetization loop detected by the MCD signal was determined by the ratio of a.c. component at 50.052 kHz (measured by a lock-in

amplifier) and a.c. component at 759 Hz (frequency of chopper, measured by a lock-in amplifier).

**Device fabrication and transport measurement**. A thick poly (methyl methacrylate) (PMMA 950, A5) film was deposited by spin coating at 4000 rpm for 60 s and then was baked at 120 °C for 60 s. Hall bar devices were fabricated by a standard electron beam lithography technique, followed by electron beam evaporation of Ti/Au (5/60 nm) electrodes. Transport measurements were performed in a PPMS (Quantum Design Inc.) cryostat. A constant 1 μA a.c. current at 17.79 Hz was applied for all the magneto-transport measurements with an SR860 lock-in amplifier.

**Density functional theory calculations**. The ab initio calculations were carried out in the framework of density functional theory (DFT) with the projector augmented wave method[42,43], as implemented in the VASP. A plane-wave basis set was used with a kinetic energy cutoff of 300 eV. The lattice constant was from our experimental observations. Atomic positions in one cubic lattice were allowed to fully relax until residual forces were less than $1 \times 10^{-3}$ eV/Å. The Monkhorst–Pack $k$ points were $15 \times 15 \times 7$ for the bulk and $15 \times 15 \times 1$ for the monolayer, and SOC was included for the self-consistent electronic calculations of MAE. The DFT + U method is used for the calculation considering the Coulomb screening[44].

## Data availability

The data that support the study are available in this paper and corresponding Supplementary Information. Additional data are available from the corresponding authors upon reasonable request.

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

## Acknowledgements

This work was supported by the National Key R & D Program of China (Grant No. 2018YFA0306900, 2018YFA0305800, 2017YFA0206301, and 2016YFA0202300), the Natural Science Foundation of China (51872012 and 51872285), and Beijing Outstanding Young Scientist Program (BJJWZYJH01201914430039). This research was supported by the high performance computing (HPC) resources at Beihang University.

## Author contributions

L.M., Z.Z., M.X., and S.Y. contributed equally to this work. Y.G. and L.M. conceived and designed the experiments and wrote the manuscript. L.M. synthesized the sample and performed majority of the materials characterization. Z.Z., L.B., H.-J.G., and L.M. designed the devices and carried out the magneto-transport measurement. M.X. and W. Z. worked on the STEM experiments and analyzed the data. S.Y. and Y.Y. measured the polar reflective magnetic circular dichroism (RMCD) of the samples. K.S., L.L., X.W., H.J., B.L., P.Z., and J.W. discussed the data. P.Q., Z.L. and L.M. performed the temperature-dependent VSM detection. P.T. carried out the theoretical calculation. All the authors participated in discussions and approved the manuscript.

## Competing interests

The authors declare no competing interests.
