## [Peer Review File · Nature Communications]

Reviewers' comments:

Reviewer #1 (Remarks to the Author):

The authors present a study of the magnetic properties of few to multilayer CrTe₂ crystallites synthesized by CVD direct on silicon dioxide layers by elemental Te and CrCl₂ precursors. They find a mean crystallite thickness dependent upon the growth temperature and time, and also present results on the degradation of such crystallites in ambient conditions. They present results on the magnetic anisotropy of crystallites (en-masse) down to 3 nm thick, a decrease of TC with increasing crystallite thickness, as well as magnetotransport measurements for samples down to 8 nm thick.

In general the article is very well written, containing excellent characterisation of the materials and their properties, and I think the results are a valuable contribution to the field. In many areas I have no comments - the STEM imaging and EELS mapping is excellent structural evidence, and the VSM and magnetotransport measurements are very clearly presented. However there are a number of other issues that require clarification.

I find it strange overall that there is no statistical presentation of the spread of flake thicknesses obtained at different growth temperatures - only in Fig1b what I assume to be the mean is presented (implying that the authors already have this data in fact). The variation of contrasts seen in S8a certainly implies a relatively large spread of values for intermediate thickness samples.

Abstract - Some acronym definitions appear here and should not.

line62

- It seems inappropriate that only two of one of the present corresponding authors manuscripts (26, 27) should be cited as general references for CVD growth of 2D materials given the extent of the literature.

line65

- "due to the uncertainty and limited characterization": what uncertainty is being referred to here?

line76

- we are missing a concise statement of what the authors have achieved in this study.

line105

- how thick are the different pictured flakes? It may not be appropriate to argue about their stability as measured by Raman when the bulk of the thin crystal is not exposed to ambient atmosphere.

lines209-218

- The lack of transport and RMCD observations of FM behaviour in samples thinner than 5.0 nm being attributed to 'degradation occur[ing] during nanofabrication' surely suggests that the crystals are not as stable as the authors suggest in the conclusion. Although the present work stands on its own, it would seem reasonable to suggest that the trend might continue for even thinner but encapsulated samples (which are becoming more routine to produce).

- In addition, 7.6 nm is the thinnest material characterised here to be precise, not 5.0 nm.

- I do not think comments about the chemical assembly of 2D ferromagnet driving future applications fit here, and are better gathered with other such statements in the manuscript conclusion.

Figure 1

- c) shows the thinnest flake presented in the manuscript at 1.2 nm step size, but there is some discrepancy. The dashed line is not normal to the flake edge, so the thickness profile shown cannot be correct, as the step edge would be broadened due to averaging. Without averaging, I find it hard to believe that the roughness can be as low as presented - perhaps 0.2\AA , estimating from the inset. This is anyway much lower than a typical silicon dioxide roughness over such a lateral scale! Perhaps the authors could show the reviewers their original data?

Figure S3

- Shows what appears to be multiple 'etch pits' with trigonal symmetry probably resulting from ambient exposure - however the authors do not comment on the origin of these etch pits in the supplementary information nor the main article, despite their similarity to those seen in Figure 1d) 4th inset.

- The sentence that follows describing the thinnest single crystal obtained at 0.6 nm is presented without corresponding evidence, and such an AFM result would also be surprising given the chemical contrast of the CrTe₂ flake vs. the substrate (see also above). I do not think the authors need to claim monolayer growth at any rate - and if such a claim should appear then it should be substantiated and in the main text.

Figure S6

- what is the reason for selecting differing flakes in a-d, rather than tracking the degradation of a single flake? Are we comparing apples and apples? How can the authors guarantee that the flakes are of the same or even of similar thickness? Especially given the already demonstrated thickness dependent Raman intensity.

- d) is clearly inhomogenous over a the few micron scale, where exactly have the Raman point spectra been acquired?

- Where is the image corresponding to the 15 day sample? Has the flake fully disappeared, or is there some residue of e.g. Cr?

The article requires proofreading:

176 slop -> slope

177 AH used without definition

189 occurred -> occurring

316 powers -> powders

345 missing symbol -> (deg)C

356 focus -> focused

371 ration -> ratio

SI117 it is not clear what is meant by 'maximum domain thickness is used to mark the samples'

SI54 optical contrast cannot become 'blurry' - the contrast is reduced, but likely there is no blurring. Dark field optical microscopy would definitely pick up any roughening of the flake boundary.

T. Booth

Reviewer #2 (Remarks to the Author):

This work reports the successful CVD growth of 1T-CrTe₂ nanosheets. The crystal structure was confirmed by STEM, while the magnetic order is studied by VSM, magneto-transport, and RMCD. Enhancement of perpendicular anisotropy (PMA) and T_c is observed with decreasing thickness, which is attributed to the weakening of the Coulomb screening in the 2D limit, based on the first principle calculation. The scientific results taken at face value are of sufficient interest. However, in my view, the authors have insufficiently accounted for potential artifacts or other possibilities that can affect this delicate conclusion, and more experimental evidence is needed to support their

interpretation. Therefore, I cannot recommend the manuscript to be published on Nature Communication before the following problems are addressed satisfactorily:

1. In the abstract, the authors claim: 'A robust anomalous Hall effect (AHE) is observed in 1T-CrTe₂, which has never been observed in other 2D materials grown by CVD', which is not appropriate. For example, clear AHE has been observed in CVD-growth FeTe nanosheets [arXiv:1912.06364,2019].
2. In figure 3e, the authors summarize the remanence and coercive force at zero fields measured by VSM, for 1T-CrTe₂ samples of various thickness. However, VSM is an average measurement of the whole substrate, and the thickness doesn't look very uniform over a wide area (figure S8a). Is the layer number accurate? What's the error bar? The authors should do statistics over a wide area.
3. I believe the standard way of determining the magnetocrystalline anisotropy of an FM is comparing the saturation fields at the well-ordered temperature, which are directly related to the anisotropy constants [Soshin Chikazumi-Physics of Ferromagnetism]. Following this way, ~40nm of CrTe₂ shows PMA as M saturate at the lower field under H//c at 100 K. This is in contradicts with the authors' conclusion of in-plane anisotropy of ~40 nm CrTe₂. What is the physical model used in the authors' analysis? How to explain the discrepancy? This affects the validity of one of the main points of this manuscript. I suggest that the authors should measure the thickness dependence of anisotropy by AHE or RMCD, where the signal to noise ratio is much better and the thickness is much more reliable.
4. In figure 3d, there are 2 slop changes at ~0.3T and ~1.2T at 100K. What's the origin of them? More careful measurement and analysis are need.
5. In figure S10, what model is used here to determine T_c? Note that 0.13 T is not a small field here, while the criticality is expected to be strongly affected by the field. Why 10 nm (figure S10 c,d) show stronger M than 40 nm? Did the authors normalize the signal by the area?
6. In figure 4g, the T_c determined from AHE is 20-30 K higher than the RMCD, and it is attributed to the laser heating during the RMCD measurement. What laser power is used in the RMCD measurement? The laser can hardly cause such a strong heating effect, especially at such high environmental temperatures (~200 K). In fact, I believe this is an indication that the T_c is not properly determined in the experiment. The reason is AHE signal is not the same as M. They are off by the anomalous Hall coefficient, which is a function a temperature, and can even change sign with temperature. [for example: PRB 98, 180408(R) (2018)]. This makes it very tricky to determine T_c from Arrott plots of AHE. The way used in Figure S16 to determine T_c from RMCD magnitudes is even more unreliable. First, it holds some problem an AHE that the coefficient between RMCD and M is a function of temperature, how did the authors account for this problem in their fitting model to determine T_c? In such a wide temperature range, how did the authors exclude the effect of position drift which can potentially affect the RMCD amplitude?
7. The author said, 'we do not observe magnetic signals from transport and RMCD measurement in samples thinner than 5.0 nm, might be due to possible degradation occurred during nanofabrication.' This looks very strange for me in several respects: 1. Did the author do any characterization to check if the sample is really degraded? If this is the true reason, then this material is not as stable as the authors claimed, which defeats one the main novelty of this paper. Moreover, the possible degradation for the thin flakes also provides an alternative explanation of the anomalous behavior of the thin flakes, challenging the validity of the authors' coulomb screening interpretation; 2. Can the authors explain what kind of nanofabrication was done for the RMCD measurement? I believe RMCD could be done without any extra nanofabrication. 3. If the sample is not degraded, it will be very strange not to see magnetic order in samples thinner than 5.0 nm in AHE and RMCD but seeing the magnetic order in samples with ~3 nm in VSM. To me, it seems to suggest the VSM data of ~3nm sample is not reliable: possibly from substrate artifact or some thick flakes (which is related to the question2).
8. There is also some problem with the theoretical interpretation. In line 166, the authors said, 'Coulomb screening is weakened in the atomic 2D thin film, resulting in a large U from the electrostatic interaction of the substrate'. I believe this is not a correct statement: since the dielectric constant of the substrate is larger than that of vacuum, the coulomb screen should be enhanced from the electrostatic interaction of the substrate, resulting in smaller U. Moreover, in figure 3e and f, the authors calculate the MAE as a function of on-site Coulomb potential with the

range of 0-8 eV to interpret the emergence of PMA (which needs to be further confirmed). However, I believe the screening of the substrate can hardly have a significant effect on ~ 7 nm metallic CrTe₂ due to such high carrier concentration.

Reviewer #3 (Remarks to the Author):

This work presents a comprehensive report on synthesis, structural characterisation and transport measurements on CVD grown CrTe₂. The authors claim observation of intrinsic ferromagnetism in the sample confirmed via robust Hall resistance measurement and demonstrating anomalous Hall effect. Layer dependent measurements show a transition in the magnetic easy axis from in-plane to out-of-plane. However, the experimental evidence for the above claims are not as convincing as one would like it to be.

Concerns:

Structural characterization using STEM-HAADF has been shown. However, the explanation on the analysis can be improved to help the readers understand the same.

XPS data shows the oxidation state of the sample however, this does not conclude on the phase of the material. Raman analysis shows 2 peaks. A mere comparison with VTe₂ is not sufficient to conclude that this is 1T phase. It might be useful to do polarization dependence and also have an idea on the number of modes to be expected in XX and XY polarization modes to verify the phase of the sample. It could be 1T, 1T' or 2H.

The VSM measurements in Fig. 3 (a) and (b) show clear MH loops with hysteresis showing magnetism. A similar measurement has been reported (arxiv:1909.09797), however no mention of this is made.

In Fig. 3(b), one can observe an additional minor step-like feature at low field (nearly 1000 Oe or so), what is the reason for this feature? It would be good to explain this.

The data in Fig. 3(c) showing magnetism has no saturation. It is necessary to measure at lower temperature or at higher fields.

Surely there is a change in the easy axis as the number of layers is decreased however, to claim a trend it is important to have more than just 2-3 data points.

The Raman spectrum as a function of number of layers has been shown; however, what is the thickness of the sample used for this? It is possible that the environmental degradation is slow for thick samples but for samples at very thin limits the degradation is quick. In such a case this can lead to spurious signals in the MH measurements and be misleading.

The reason for increase in T_c with decrease in number of layers is not convincing. A more detailed explanation is required to understand this.

To show anomalous Hall effect magnetotransport and RMCD have been used. However, it is not clear why RMCD is necessary. This does not seem to give any additional information. It can be moved to supplementary.

At what temperature is the DFT calculation for MAE accounted for? Can you show the MAE at which the anisotropy is observed significantly?

The authors claim applications based on ferromagnetism - while the mobility is too low, this is highly unlikely. It is not clear that this is more advantageous over CVT or other methods. A comparison has to be clearly established with CVT and MBE grown samples to make such a claim. With the above concerns and prior work in this field, this work is more suitable for a journal like Scientific Reports or Phys. Rev. B after incorporating the corrections as suggested. This is certainly an interesting incremental work presenting details of synthesis of another 2D material to be added to the shelf.

We thank the referees for the helpful and positive comments and have revised the paper accordingly to address the points raised. In this point-to-point response letter, comments from the referees are in black typeface, and our responses are in the blue typeface. Major changes have been highlighted in blue in the revised main text and the supplementary information.

Reviewers' comments:

Reviewer #1 (Remarks to the Author):

The authors present a study of the magnetic properties of few to multilayer CrTe₂ crystallites synthesized by CVD direct on silicon dioxide layers by elemental Te and CrCl₂ precursors. They find a mean crystallite thickness dependent upon the growth temperature and time, and also present results on the degradation of such crystallites in ambient conditions. They present results on the magnetic anisotropy of crystallites (en-masse) down to 3 nm thick, a decrease of T_C with increasing crystallite thickness, as well as magnetotransport measurements for samples down to 8 nm thick.

In general the article is very well written, containing excellent characterisation of the materials and their properties, and I think the results are a valuable contribution to the field. In many areas I have no comments - the STEM imaging and EELS mapping is excellent structural evidence, and the VSM and magnetotransport measurements are very clearly presented. However there are a number of other issues that require clarification.

We thank Reviewer#1 very much for reading our manuscript carefully and providing valuable comments and suggestions, which help us to improve the quality of the work. We have provided a detailed point-to-point response to address the following concerns.

1. I find it strange overall that there is no statistical presentation of the spread of flake thicknesses obtained at different growth temperatures - only in Fig1b what I assume to be the mean is presented (implying that the authors already have this data in fact). The variation of contrasts seen in S8a certainly implies a relatively large spread of values for intermediate thickness samples.

We thank Reviewer #1 for the helpful comments. We agree with the referee that the average flake thicknesses are shown in the manuscript, and the statistical thickness that varies with growth temperature is not presented. In the revised version, we have summarized the AFM data of 1T-CrTe₂ samples obtained at different growth temperatures, and added the statistical data to Figure 1b of the revised manuscript and supporting information (SI). The result of each statistical diagram is calculated from

five batches of growth at the same temperature. As shown in Figure R1, the thickness of the resulting sample increases with increasing temperatures. It is worth noting that the sample thickness is very sensitive to temperature changes (especially at relatively high growth temperatures). Even though it is difficult to synthesize samples with precise thickness, under our optimized growth condition, the resulting nanoflakes within a certain narrow thickness range still play a leading role. This trend indicates that it is feasible to obtain a relatively narrow thickness distribution of the synthesized nanoflakes. We added Figure R1 as the Supplementary Figure 3 in the revised SI and briefly discussed the results in the revised main text.

Figure R1. The thickness distribution histograms of 1T-CrTe₂ crystals. (a - g) The samples were synthesized at the temperature of 973, 976, 983, 985, 988, 991, and 993 K, respectively. (h) The sample thickness as a function of the growth temperature. The black line and the black rectangle indicate the range of the thickness and the averaged thickness of the samples grown at a given temperature, respectively.

2. Abstract - Some acronym definitions appear here and should not.

We thank Reviewer #1 for his/her valuable suggestions. We have removed the acronym in the abstract, and their full names are used in the abstract in the revised manuscript.

3. line62

- It seems inappropriate that only two of one of the present corresponding authors manuscripts (26, 27) should be cited as general references for CVD growth of 2D materials given the extent of the literature.

We thank the Reviewer #1 for the kind suggestion. Following the reviewer's suggestion, we deleted one of the two references here, and have included more citations for the growth of 2D materials from different research groups, which includes [Nature 556, 355–359 (2018)], [Nat. Commun. 10, 2957 (2019)] and [Preprint at <https://arxiv.org/abs/1912.06364> (2019)].

4. line65

- "due to the uncertainty and limited characterization": what uncertainty is being referred to here?

We thank Reviewer #1 for his/her helpful comments. The “uncertainty” in this sentence is being referred to as the mechanisms of controlled synthesis of magnetic layered materials based on CVD. Although some inspiring works (such as Ref. 17, Ref. 30 – 33 in the revised manuscript) focused on the synthesis of layered and/or non-layered ultrathin magnetic materials, there are relatively few reports on the controllable synthesis of 2D layered materials by CVD, and the underlying growth and/or regulation mechanisms is still unclear.

To avoid the misleading, we rewrite the sentence in the revised manuscript as: *“due to the uncertainty of growth mechanism of 2D layered magnetic materials and the limited characterization of their magnetic properties, controllable synthesis of 2D magnetic materials via CVD remains a cutting-edge topic.”*

5. line76

- we are missing a concise statement of what the authors have achieved in this study.

We thank Reviewer#1 for pointing out this. We have added a concise conclusion in the revised manuscript at the end of line 76 as a new paragraph:

“In this work, we developed a CVD strategy to synthesize 1T-CrTe₂ on SiO₂/Si substrates with controlled thickness by controlling the growth temperature and atmospheric condition. Robust anomalous Hall effect (AHE) is observed in the resulting samples without any encapsulation, indicating its FM properties and good stability. Furthermore, as the thickness of 1T-CrTe₂ reducing from tens of nanometers to several nanometers, the easy axis changes from in-plane to out-of-plane, and a monotonic increase of Curie temperature is observed. Theoretical calculations indicate that the Coulomb screening plays a crucial role in the change of magnetic properties.”

6. line105

- how thick are the different pictured flakes? It may not be appropriate to argue about their stability as measured by Raman when the bulk of the thin crystal is not exposed to ambient atmosphere.

We thank Referee #1 for his/her helpful comment. Actually, in the previous version, the stability of 1T-CrTe₂ was studied by exposing the same sample (with a thickness of about 5 nm) to the ambient atmosphere for different times. To give a clear illustration, we retested a sample with a thickness of about 4.3 nm, as shown in Figure R2 (a).

We agree with Reviewer #1 that the Raman spectroscopy is a surface-sensitive test method, however, for samples with a thickness of only a few nanometers, the Raman signal has the ability to reflect the material information [Phys. Rev. B. 87, 195316

(2013)]. Thus, Raman spectroscopy is widely used in the investigations of 2D materials. Alternatively, RMCD is also a non-destructive method for detecting magnetic materials. To provide more evidence that the sample has degraded or not, here we conducted RMCD tests under different air exposure time. As shown in Figure R2 (b), the almost unchanged RMCD signals after 5 days of exposure in the air indicate that our sample has good environmental stability. After 15 days of exposure to air, no RMCD signals are detected, indicating that the sample has degenerated. The results of RMCD are comparable to that of the Raman test. In the revised manuscript, we added this evidence as Supplementary Figure 7 in the SI.

Figure R2. AFM and RMCD data of 1T-CrTe₂ for environmental stability investigations. (a) AFM image and height profile of the sample being tested. (b) RMCD signal as a function of exposure time under ambient condition.

7. lines209-218

- The lack of transport and RMCD observations of FM behaviour in samples thinner than 5.0 nm being attributed to 'degradation occur[ing] during nanofabrication' surely suggests that the crystals are not as stable as the authors suggest in the conclusion. Although the present work stands on its own, it would seem reasonable to suggest that the trend might continue for even thinner but encapsulated samples (which are becoming more routine to produce).

- In addition, 7.6 nm is the thinnest material characterised here to be precise, not 5.0 nm.

- I do not think comments about the chemical assembly of 2D ferromagnet driving future applications fit here, and are better gathered with other such statements in the manuscript conclusion.

We greatly appreciate Reviewer #1 for careful review and advice. In the revised manuscript, we investigated the environmental stability of a few-layered 1T-CrTe₂ sample (4.3 nm), as shown in Figure R2 (b). According to the RMCD results, the magnetic properties of the tested sample can be maintained for more than five days. In contrast, the bulk or few-layered CrTe₂ synthesized indirectly by oxidation of KCrTe₂ is

easily degraded in air and must be covered with hexagonal boron nitride (h-BN) in a glove box for further characterization [Preprint at <https://arxiv.org/abs/1909.09797> (2019)]. It is worth noting that as the thickness decreases, other layered magnetic samples become extremely unstable [Nature 546, 265-269 (2017)]. Thus, our few-layered samples grown by CVD show higher environmental stability than that CrTe₂ obtained by indirect oxidation. Meanwhile, we tested two other samples about 7.5 nm and 9 nm thick to further illustrate the trend of T_c , as shown in Figure R3. In the revised manuscript, we added this evidence as Supplementary Figure 17 in the SI. We changed the conclusion “*The processability in air and the observation of robust AHE without any encapsulation prove the excellent stability of this material*” to “*The processability in air and the observed robust AHE without any encapsulation for the samples with a thickness greater than 5 nm prove the relatively good stability of this material*”

We agree with Reviewer #1 that it seems reasonable to suggest that the trend might continue for even thinner but encapsulated samples, and we will continue to work on it for further study.

We thank Referee #1 for pointing out our misleading expression. The thinnest thickness of the measured sample is 4.3 nm, which is clearly stated in the revised manuscript. We have updated the data and corrected the expression in the revised manuscript.

As suggested by the reviewer, we revised the sentence to read: “*Nevertheless, the chemically assembled large-scale 2D ferromagnets with higher T_c in thinner crystals (determined by three different characterization methods) is interesting, but the further detailed study is needed.*”

Figure R3. RMCD measurements of 1T-CrTe₂ with thicknesses of about 7.5 nm (a) and 9.3 nm (b) under several given temperatures, respectively.

8. Figure 1

- c) shows the thinnest flake presented in the manuscript at 1.2 nm step size, but there is some discrepancy. The dashed line is not normal to the flake edge, so the thickness profile shown cannot be correct, as the step edge would be broadened due to averaging.

Without averaging, I find it hard to believe that the roughness can be as low as presented - perhaps 0.2\AA , estimating from the inset. This is anyway much lower than a typical silicon dioxide roughness over such a lateral scale! Perhaps the authors could show the reviewers their original data?

We thank Reviewer #1 for pointing out this mistake. To obtain the correct thickness information, the height line profile direction should be normal to the edge of the flake. We agree with Reviewer #1's opinion that we smoothed the AFM image when processed the data. Based on the reviewer's suggestions, we corrected these mistakes and showed the original thickness line profile (raw data without smoothing) in the revised main text (Fig. 1 (c)), as shown in Figure R4.

Figure R4. AFM image and the corresponding OM image (upper right inset) of a typical 1T-CrTe₂ hexagonal nanoflake on a 285 nm SiO₂/Si substrate. The height profile in the lower left inset shows the thickness of the synthesized sample is ~ 1.2 nm.

9. Figure S3

- Shows what appears to be multiple 'etch pits' with trigonal symmetry probably resulting from ambient exposure - however the authors do not comment on the origin of these etch pits in the supplementary information nor the main article, despite their similarity to those seen in Figure 1d) 4th inset.

- The sentence that follows describing the thinnest single crystal obtained at 0.6 nm is presented without corresponding evidence, and such an AFM result would also be surprising given the chemical contrast of the CrTe₂ flake vs. the substrate (see also above). I do not think the authors need to claim monolayer growth at any rate - and if such a claim should appear then it should be substantiated and in the main text.

In our previous manuscript, we tried to give two aspects of information from Figure S3, the first is the layered nature of 1T-CrTe₂, and the other is the thickness of the single-layer 1T-CrTe₂ crystal. Meanwhile, AFM thickness data and cross-sectional STEM-HAADF data can be mutually verified. We found that the top layer of some

samples was not completely covered, which has also been observed in the growth of other 2D materials, resulting in pits on the surface of the synthesized nanoflakes [ACS Nano 13, 3649-3658 (2019); Nature Mater. 12, 754-759 (2013)]. The origin of these multiple pits with trigonal symmetry may probably be caused by the layer-by-layer growth mechanism. Note that the multiple pits in Figure S3 are very similar to those in Figure 1d, and it is reasonable to infer that the pits in Figure 1d may be caused by layer-by-layer degradation [ACS Appl. Mater. Interfaces. 12, 30702 (2020)].

The thickness of single-layer 1T-CrTe₂ extracted from the AFM height profile is about 0.62 nm, which is comparable to that from cross-sectional STEM-HAADF data and literature reports [J. Phys.: Condens. Matter. 27, 176002 (2015)]. The AFM data improves our confidence in inferring that the 1T-CrTe₂ has a layered structure. As suggested by the reviewer, we added the discussion to Figure S3 (Supplementary Figure 4 in the revised manuscript) and highlighted it in blue in the revised SI.

We agree with Reviewer #1 that we don't need to claim monolayer growth at any rate. Therefore, we changed "*The thinnest step in the single crystal we obtained is 0.6 nm (Supplementary Fig. 4), which is in good accordance with the thickness of the monolayer 1T-CrTe₂*" to "*A step height of about 0.6 nm can be observed in some samples (Supplementary Fig. 4), confirming the layered nature of 1T-CrTe₂.*"

10. Figure S6

- what is the reason for selecting differing flakes in a-d, rather than tracking the degradation of a single flake? Are we comparing apples and apples? How can the authors guarantee that the flakes are of the same or even of similar thickness? Especially given the already demonstrated thickness dependent Raman intensity.

- d) is clearly inhomogenous over a the few micron scale, where exactly have the Raman point spectra been acquired?

- Where is the image corresponding to the 15 day sample? Has the flake fully disappeared, or is there some residue of e.g. Cr?

We greatly appreciate Reviewer #1 for careful review. Actually, the environmental stability investigations of 1T-CrTe₂ are carried out in the same sample. The discrepancy in OM results may originate from the images obtained on different microscopes. To provide a clear illustration, we retested a sample with a thickness of about 4.3 nm, as shown in Figure R5 (Supplementary Figure 7 in the revised manuscript). Figure R5 shows the OM images and corresponding Raman signals of 1T-CrTe₂ samples exposed to the atmosphere for 0 day, 5 days, 7 days, 9 days, 10 days, 12 days, 14 days, and 15 days, respectively. After being exposed to the air for 5 days, the optical contrast and morphology of the sample did not change, and the Raman intensity and RMCD signal were comparable to those of the fresh one. When the exposure time is extended to more than 10 days, the optical contrast and morphology of 1T-CrTe₂ become lighter and

rugged. The corresponding Raman signal intensities decrease and eventually vanish. Finally, after 15 days of exposure to the air, no RMCD and Raman signals are detected, indicating that the sample has degraded. In the revised manuscript, we added these evidences as Supplementary Figure 7 in the SI.

Ambient temperature and humidity conditions may have a significant impact on the stability investigations, thus we have also recorded the data simultaneously and added these data in the revised manuscript (Supplementary Figure 7).

Figure R5. Environmental stability investigations. (a) Optical images of 1T-CrTe₂ samples exposed in the atmosphere for 0 day, 5 days, 7 days, 9 days, 10 days, 12 days, 14 days and 15 days, respectively. (b) AFM image of the sample being tested. (c) The corresponding Raman spectra from a. (d) Humidity and temperature data during the test.

11. The article requires proofreading:

176 slop -> slope

177 AH used without definition

189 occurred -> occurring

316 powers -> powders

345 missing symbol -> (deg)C

356 focus -> focused

371 ration -> ratio

We are very grateful for Reviewer #1's reminder about the grammar and spelling mistakes, and we carefully proofread our manuscript and corrected the grammar and spelling mistakes thoroughly.

12. SI117 it is not clear what is meant by 'maximum domain thickness is used to mark the samples'

We thank Reviewer #1 for pointing out this. It is worth noting that the VSM test can only obtain the averaged signal, and the thickness of the sample will significantly affect the signal obtained. Before processing the VSM test, we statistically analyzed the thickness data of 1T-CrTe₂ samples on a large scale and used the dominant sample thickness (the sample thickness with the highest percentage) to name the samples. In the revised manuscript, we changed "*The maximum domain thickness is used to mark the samples because the average signal was obtained when using VSM as the measurement method.*" to "*Before processing the VSM/SQUID test, we statistically analyze the thickness data of 1T-CrTe₂ samples on a large scale, and use the sample thickness with the dominate percentage to name the sample.*"

13. SI54 optical contrast cannot become 'blurry' - the contrast is reduced, but likely there is no blurring. Dark field optical microscopy would definitely pick up any roughening of the flake boundary.

We thank Reviewer #1 for this useful comment. As advised by the reviewer, we changed the description in the revised manuscript as: "*As shown, the optical contrast is reduced when the hydrogen concentration increases.*"

Reviewer #2 (Remarks to the Author):

This work reports the successful CVD growth of 1T-CrTe₂ nanosheets. The crystal structure was confirmed by STEM, while the magnetic order is studied by VSM, magneto-transport, and RMCD. Enhancement of perpendicular anisotropy (PMA) and T_c is observed with decreasing thickness, which is attributed to the weakening of the Coulomb screening in the 2D limit, based on the first principle calculation. The scientific results taken at face value are of sufficient interest. However, in my view, the authors have insufficiently accounted for potential artifacts or other possibilities that can affect this delicate conclusion, and more experimental evidence is needed to support their interpretation. Therefore, I cannot recommend the manuscript to be published on Nature Communication before the following problems are addressed satisfactorily:

We thank Referee #2 for reading our manuscript carefully and providing valuable comments and advice, which help us to improve the quality of this paper. We have provided a detailed point-to-point response to address the following concerns.

1. In the abstract, the authors claim: ‘A robust anomalous Hall effect (AHE) is observed in 1T-CrTe₂, which has never been observed in other 2D materials grown by CVD’, which is not appropriate. For example, clear AHE has been observed in CVD-growth FeTe nanosheets [arXiv:1912.06364, 2019].

We thank Referee #2 for this helpful comment. As advised by Reviewer #2, we carefully read this paper. Kang and coworkers reported a CVD-based rational growth approach for the synthesis of ultrathin FeTe crystals with controlled structural and magnetic phases. FeTe nanoplates with either layered tetragonal or non-layered hexagonal phase can be controlled by precisely optimizing the growth temperature. Transport measurements reveal that layered tetragonal FeTe is antiferromagnetic with a Néel temperature of about 71.8 K, while non-layered hexagonal FeTe is ferromagnetic with a Curie temperature around 220 K.

We agree with Reviewer #2 that Kang and coworkers have observed a clear AHE in CVD grown non-layered FeTe nanosheets. Our manuscript reports AHE observed in CVD grown 2D layered materials. To give a more careful statement, we removed the claim “*which has never been observed in other 2D materials grown by chemical vapor deposition.*” and revised it as “*which has seldom been observed in other layered 2D materials grown by chemical vapor deposition.*”

2. In figure 3e, the authors summarize the remanence and coercive force at zero fields measured by VSM, for 1T-CrTe₂ samples of various thickness. However, VSM is an average measurement of the whole substrate, and the thickness doesn’t look very uniform over a wide area (figure S8a). Is the layer number accurate? What’s the error bar? The authors should do statistics over a wide area.

We thank Reviewer #2 for his/her helpful comments. We agree with the reviewer's comment that the VSM measurements obtain averaged signals and the statistical thicknesses study as a function of growth temperature is not presented in the manuscript. In the revised version, we have summarized the AFM data of 1T-CrTe₂ samples obtained at different growth temperatures, and added the statistical data with error bars to Figure 1b of the revised manuscript and supporting information (SI). The result of each statistical diagram is calculated from five batches of growth at the same temperature. As shown in Figure R6, the thickness of the resulting sample increases with increasing temperatures. It is worth noting that the sample thickness is very sensitive to temperature changes (especially at relatively high growth temperatures). Even though it is difficult to synthesize samples with precise thickness, under our optimized growth condition, the resulting nanoflakes within a certain narrow thickness range still play a leading role. This trend indicates that it is feasible to obtain a relatively narrow thickness distribution of the synthesized nanoflakes. Thus, VSM measurement can be roughly used to show the thickness-dependent magnetic properties. We added Figure R6 as the Supplementary Figure 3 in the revised SI and briefly discussed the results in the revised main text.

As advised by the reviewer, we re-grew some samples and measured the remanence and coercive force at zero fields by SQUID, as shown in Figure R7. The thickness of the samples we selected is about 3, 5, 10, 35, and 40 nm, respectively. Meanwhile, we normalized the remanence signal by the sample area (which is related to question 5). We replaced Figure 3e with Figure R7 in the revised manuscript.

Figure R6. The thickness histogram distributions of 1T-CrTe₂ crystals. (a - g) The samples were synthesized at the temperature of 973, 976, 983, 985, 988, 991, and 993 K, respectively. (h) The sample thickness as a function of growth temperature. The black line and the black rectangle indicate the range of the thickness and the averaged thickness of the samples grown at a given temperature, respectively.

Figure R7. Remanence and coercive force at zero field for 1T-CrTe₂ samples of various thicknesses. The magnetic hysteresis loops at 100 K was used to extract the value of remanence and coercive force.

3. I believe the standard way of determining the magnetocrystalline anisotropy of an FM is comparing the saturation fields at the well-ordered temperature, which are directly related to the anisotropy constants [Soshin Chikazumi-Physics of Ferromagnetism]. Following this way, ~40nm of CrTe₂ shows PMA as M saturate at the lower field under H//c at 100 K. This is in contradicts with the authors' conclusion of in-plane anisotropy of ~40 nm CrTe₂. What is the physical model used in the authors' analysis? How to explain the discrepancy? This affects the validity of one of the main points of this manuscript. I suggest that the authors should measure the thickness dependence of anisotropy by AHE or RMCD, where the signal to noise ratio is much better and the thickness is much more reliable.

We thank Reviewer #2 for providing these comments. We agree with Reviewer #2 for the way of determining the magnetocrystalline anisotropy. It is inappropriate to determine the easy axis only by the comparison of the remanent magnetism in the original manuscript.

In the revised manuscript, we use SQUID to retest the samples with a thickness of about 40 nm, which is considered to have a higher resolution than that of VSM [Mater. Today 27, 107 (2019)]. The RMCD or AHE should have been used to determine the thickness dependence of magnetic anisotropy, but due to the equipment limitations (they only provide magnetic field along the c-axis), we cannot achieve such measurements. As shown in Figure R8, 40 nm thick 1T-CrTe₂ shows PMA as M saturate at the lower field under H ⊥ c-axis at 100 K, indicative of an in-plane easy-axis for thick 1T-CrTe₂ crystals. Meanwhile, comparing magnetic moment values in different directions can also be used to estimate the easy axis [Adv. Funct. Mater. 30, 1910036 (2020)]. Following this way, 40 nm thick 1T-CrTe₂ shows a higher magnetic moment in H ⊥ c-axis compare to that of H // c-axis, indicative of an in-plane easy-axis. We replaced

Figure 3 (a, b) with Figure R8 (a, b) in the revised manuscript and added Figure R9 to SI.

Figure R8. (a, b) Magnetic hysteresis loops for 1T-CrTe₂ flakes with a thickness of ~ 40.0 nm under the magnetic field parallel (a) and vertical (b) to the c-axis of the crystal, respectively.

Figure R9. M-T curve of a 40 nm thick 1T-CrTe₂ sample.

4. In figure 3d, there are 2 slope changes at ~0.3T and ~1.2T at 100K. What's the origin of them? More careful measurement and analysis are need.

We thank Reviewer #2 to point out this issue. To make it clear, we retest the magnetic properties for 3 nm thick sample by SQUID and the result is shown in Figure R10. After comparing the data, we speculate that the slope changes at ~1.2 T may originate from the low signal-to-noise ratio when performing VSM measurements. The diamagnetic background from the substrate provides more signals and finally results in the slope changes. Comparably, as shown in Figure R10(b), the slope changes at ~0 T may be caused by the weak magnetic signal from the ultrathin samples [Preprint at <https://arxiv.org/abs/1912.06364> (2019)]. We changed Figure 3d with Figure R10 and added “The slope changes at ~0 T under a relatively lower temperature may be caused by the weak magnetic signal from the ultrathin samples.” in the revised manuscript.

Figure R10. Magnetic hysteresis loops for 1T-CrTe₂ flakes with a thickness of ~ 3.0 nm under the magnetic field vertical to the c-axis of the crystal. b is an enlarged view of a.

5. In figure S10, what model is used here to determine T_c ? Note that 0.13 T is not a small field here, while the criticality is expected to be strongly affected by the field. Why 10 nm (figure S10 c,d) show stronger M than 40 nm? Did the authors normalize the signal by the area?

We thank Reviewer #2 to point out this issue. By following previous studies [Dalton Trans. 35, 4708 (2008); Adv. Funct. Mater. 30, 1910036 (2020)], we determine the T_c via the extraction of the inflection point defined by the minimum of dM/dT in the $M-T$ curve, as shown in Figure R11. We retested the samples with various thicknesses and updated the data as Supplementary Figure 11 and 12 in the revised manuscript. The method to determine T_c was added to the revised manuscript as “*The T_c was extracted from the inflection point defined by the minimum of dM/dT in the $M-T$ curve.*”

In the previous version, we did not normalize the magnetic signal by area and the discrepancy in M may cause by the different coverages of the synthesized samples on the substrate, which is a common phenomenon in samples grown by CVD [J. Am. Chem. Soc. 140, 14217-14223 (2018)]. As advised by the reviewer, the magnetic moment is normalized by the area and the results are shown in Figure R11 (a-c). We updated the data as Supplementary Figure 11 in the revised manuscript.

We agree with Reviewer #2’s comments that the magnetic field has a significant effect on the criticality and a proper magnetic field is very important for determining the T_c . Thus, we compared the resulted $M-T$ curve under different magnetic fields. Meanwhile, the magnetic moments from in-plane and out-of-plane have also been compared. Noting that the magnetic moment is normalized by the area. As shown in Figure R12, the magnetic moment increases firstly and then decreases as the magnetic field increases at the same temperature. The extracted T_c at both in-plane and out-of-plane magnetic field under a low magnetic field is slightly lower than that under a high magnetic field. Thus, by comparing the values of magnetic moment and T_c under the

three magnetic fields, we infer that 1300 Oe may be a suitable magnetic field to perform the measurements (M - T curve).

Figure R11. M - T curve of 1T-CrTe₂ with a thickness of about 40 nm (a), 10 nm (b) and, 3 nm (c), respectively. The corresponding dM/dT under the magnetic field vertical and parallel to the c -axis of the crystal are shown in (d - f).

Figure R12. M - T curve of 1T-CrTe₂ with a thickness of ~ 40.0 nm under the magnetic field vertical (a) and parallel (b) to the c -axis of the crystal, respectively.

6. In figure 4g, the T_c determined from AHE is 20-30 K higher than the RMCD, and it is attributed to the laser heating during the RMCD measurement. What laser power is used in the RMCD measurement? The laser can hardly cause such a strong heating effect, especially at such high environmental temperatures (~ 200 K). In fact, I believe this is an indication that the T_c is not properly determined in the experiment. The reason is AHE signal is not the same as M . They are off by the anomalous Hall coefficient, which is a function a temperature, and can even change sign with temperature. [for example: PRB 98, 180408(R) (2018)]. This makes it very tricky to determine T_c from Arrott plots of AHE. The way used in Figure S16 to determine T_c from RMCD

magnitudes is even more unreliable. First, it holds some problem an AHE that the coefficient between RMCD and M is a function of temperature, how did the authors account for this problem in their fitting model to determine T_c ? In such a wide temperature range, how did the authors exclude the effect of position drift which can potentially affect the RMCD amplitude?

We thank Reviewer #2 for his/her careful review and the comments are constructive. The laser power of $0.36 \mu\text{W}$ is used during the RMCD measurement, and the laser spot is focused to be about $2 \mu\text{m}$ in diameter. The reason for the discrepancy in T_c extracted from AHE and RMCD may be complicated and need further exploration. The AHE measurements are performed on the fabricated 1T-CrTe₂ nanoflakes, while RMCD measurements are performed on the as-grown ones. Firstly, the local laser heating during RMCD measurement may cause a signal disturbance to some extent, thus affect the final results. Secondly, the RMCD signal is sensitive to the out-of-plane magnetization, the magnetic anisotropy changes in 1T-CrTe₂ will have an impact on the measurement of T_c [Nature Nanotech. 13, 544–548 (2018)]. Thirdly, the existence of domains wall may induce a discrepancy in the T_c extracted from AHE and RMCD [Nature 563, 94-99 (2018)]. The last not the least, T_c is very sensitive to the electron density, the potential doping from metal contact (electrode) or organic/inorganic contamination during the nanofabrication will induce a large influence [Nature Nanotech 14, 408–419(2019)]. Considering those influences, it is difficult to establish the origin of the discrepancy, the reason for the discrepancy in T_c may need further exploration. In the revised manuscript, we added “*Note that the estimated T_c from AHE is consistently higher than the results from RMCD (Fig. 4g), which may result from many reasons, such as the laser heating during RMCD measurement, the existence of domains wall and the doping effect during the nanofabrication. Considering the complexity of the reason for the discrepancy in T_c extracted from AHE and RMCD, further efforts are needed to unravel the fundamental issue.*” in the main text.

As advised by Reviewer #2, we carefully read and analyzed these insightful works. We agree with Reviewer #2’s opinion that the AHE signal is not exactly equal to M and is affected by the anomalous Hall coefficient. Thus, it is difficult to obtain the T_c exactly. Even though the extraction of T_c from the AHE signal is tricky, we can also define the T_c as the temperature at which the AHE vanishes [Phys. Rev. B 98, 180408(R) (2018); Nature 563, 94-99 (2018)]. Meanwhile, the T_c can be identified by the slight change in each R_{xx} - T curve [Nature Mater. 17, 778–782 (2018)], as shown in Figure R13. R_{xx} is the longitudinal electrical resistivity. As shown in Figure R13, the estimated T_c of 10 nm and 30 nm thick sample is about 210 K and 186 K, respectively. The result is consistent with that obtained by Arrott plots. Meanwhile, Arrott plots can be used to determine the T_c when performing the investigations in 2D magnetic materials [Nature 563, 94-99 (2018); Nature 408, 944-946 (2000)] and be able to minimize the effect of domain rotation and magnetic anisotropy [Nature 408, 944–946 (2000)]. Following this way, we agree with the reviewer that Arrott plots are not absolutely accurate to determine the T_c , but it can provide a reliable T_c within the existing technologies. We

added Figure R13 to SI as Supplementary Figure 14 in the revised manuscript.

RMCD has been proven to be a nondestructive and effective technique for probing the 2D magnetism [Nature 563, 94-99 (2018); Nature Materials 17, 778-782 (2018)]. As we know, critical exponents governing the behavior of magnetization (M) as a function of the reduced temperature, $t = T/T_c - 1$, or the magnetic field B [Nature Nanotech 14, 408-419(2019)]. Whereas the critical exponents for the materials with different modes can be determined analytically, the power-law $M(T) = M(0)(1-T/T_c)^\beta$ can also be used to extract the T_c [Adv. Funct. Mater. 30, 1910036 (2020); Nature Materials 17, 778-782 (2018); Nature Nanotech. 14, 408-419 (2019)]. Note the criticality fits are only accurate in the vicinity of the critical point, where the correlation length diverges. When we perform the fits, we start by fitting data very close to the Curie temperature and then progressively include lower temperature data until the data are all added. This cutoff ends up being $1 - T/T_c \lesssim 0.25$. Even though the extracted T_c has a discrepancy between AHE and RMCD, the tendency of T_c as a function of thickness changes is consistent, which is similar to the observations in Fe_3GeTe_2 [Nature 563, 94-99 (2018)].

To exclude the effect of position drift, we first record the position of the samples to be measured. Then, we refocus and move the stage to the same position to measure every time after changing the temperature. Those operations can exclude the effect of position drift and ensure that the measurements under different temperatures are all in the same position.

Figure R13. Hall resistance as a function of temperature, measured on a 10.0 nm and 30 nm thick device, respectively.

7. The author said, ‘we do not observe magnetic signals from transport and RMCD measurement in samples thinner than 5.0 nm, might be due to possible degradation occurred during nanofabrication.’ This looks very strange for me in several respects:

Did the author do any characterization to check if the sample is really degraded? If this is the true reason, then this material is not as stable as the authors claimed, which defeats one of the main novelties of this paper. Moreover, the possible degradation for the thin flakes also provides an alternative explanation of the anomalous behavior of the

thin flakes, challenging the validity of the authors' coulomb screening interpretation;

Can the authors explain what kind of nanofabrication was done for the RMCD measurement? I believe RMCD could be done without any extra nanofabrication.

If the sample is not degraded, it will be very strange not to see magnetic order in samples thinner than 5.0 nm in AHE and RMCD but seeing the magnetic order in samples with ~3 nm in VSM. To me, it seems to suggest the VSM data of ~3nm sample is not reliable: possibly from substrate artifact or some thick flakes (which is related to the question2).

We thank Reviewer #2 for his/her review and correction. In our previous version, some very thin samples (such as samples with 1.2 nm thick) can hardly be found after the standard electron beam lithography process. Meanwhile, we have tried to measure the Raman spectra on surviving samples after completing the Hall tests, no signals were found. Thus, we inferred that the sample has degraded, however, no test was done after RMCD measurement.

In the revised manuscript, to figure out the degradation process in thin samples, we firstly investigated a sample of about 4.3 nm thick by RMCD and Raman spectroscopy (Fig. 1d and Supplementary Figure 7 in the revised manuscript). From RMCD results, the fresh 1T-CrTe₂ flake exhibits a rectangular hysteresis loop at 2 K, indicative of ferromagnetic ordering, as shown in Figure R14. The nearly unchanged RMCD signals after 5 days of air-exposure surely suggested that the sample has good environmental stability. After 15 days of air-exposure, no RMCD signals are detected, which indicated that the sample has degraded. However, for even thinner samples (such as 1.2 nm thick), the degradation process becomes faster and no signal can be found under RMCD, as shown in Figure R15. Even though the 1.2 nm thick samples degrade faster than the thick one, our CVD synthesized samples still have better stability than other FM layered materials.

We agree with the reviewer that we cannot completely exclude the possibilities that the degradation of the thin flakes also provides an alternative explanation of the anomalous behavior of the thin flakes. However, in references, the degradation of samples usually causes the disappearance of magnetic properties in ultrathin layers. The coulomb screening interpretation is the most possible explanation. In the future, we will conduct more experiments to validate the explanation.

We agree with Reviewer #2 that RMCD measurement does not require any nanofabrication process. In the revised manuscript, we revised the claim as *“Meanwhile, we do not observe magnetic signals from transport measurement in samples thinner than ~ 10.0 nm, which might be due to possible degradation occurred during nanofabrication.”*

To avoid the possible degradation during the SQUID or RMCD measurements in ultrathin samples (such as 3 nm thick samples), we spin-coated PMMA on the resulted samples. However, we cannot achieve such the RMCD test in ultrathin samples because of the low optical contrast of our setup where the samples thinner than 4 nm with PMMA coating cannot be recognized. Thus, the thinnest sample thickness from our RMCD measurement is ~ 4.3 nm. The results showed above may offer evidence for observing the magnetic order in samples with ~ 3 nm in SQUID. In the future, we will conduct more experiments to validate the explanation. We added the following sentence to the main text: “To avoid the possible degradation of the samples, PMMA coating is used before the VSM/SQUID test.”

Figure R14. AFM and RMCD date of 1T-CrTe₂ being investigated. (a) AFM image and height profile of the sample being tested. (b) RMCD signal as a function of exposure time under ambient condition.

Figure R15. OM and corresponding Raman signals of a 1.2 nm thick 1T-CrTe₂ nanoflake exposed to the air with the extension of time.

8. There is also some problem with the theoretical interpretation. In line 166, the authors said, ‘Coulomb screening is weakened in the atomic 2D thin film, resulting in a large U from the electrostatic interaction of the substrate’. I believe this is not a correct statement: since the dielectric constant of the substrate is larger than that of vacuum, the coulomb screen should be enhanced from the electrostatic interaction of the substrate, resulting in smaller U.

We thank Reviewer #2 to raise this issue. We believe that Referee #2 misunderstood our expressions. In our previous manuscript, we claimed that “*The effective Coulomb screening could be influenced by the dimension of the samples. Compared with that in 3D bulk, the Coulomb screening is weakened in the atomic 2D thin film, resulting in a large U from the electrostatic interaction of the substrate, finally, it is possible to flip the easy axis.*” When we claimed that “the weakening of Coulomb screening”, we were comparing the 2D atomic thin film with 3D metallic compound, but not to compare the atomic 2D thin films with and without substrate.

The on-site Coulomb interaction U on the atom with the magnetic moment in 3D and 2D materials is determined by two factors: 1) the atomic Coulomb interaction U_0 from on-site interaction of two electrons on the same d orbital; 2) the screening effect from the neighboring electron cloud around magnetic atoms. U_0 is determined by the element, but the screening effect is determined by electronic structures of the materials and strongly depends on the dimension of materials [Quantum Theory of the Optical and Electronic Properties of Semiconductor, 5 Ed, H. Haug, and S. W. Koch, World Scientific; Many-Particle Physics, 3 Ed, G. D. Mahan, Springer]. In a metallic system, the U has the form of

$$U = v(\mathbf{q})\varepsilon^{-1} + \frac{1}{2}U_0\chi^{-1}$$

In which, $v(\mathbf{q})$ is the Fourier transforming of Coulomb interaction in the momentum space. ε is the dielectric functional, which is velocity dependent and determined by the intrinsic dielectric constant (we can include the substrate influence here) and the Coulomb interaction, χ is the paramagnetic screening functional, which is also a velocity-dependent operator and determined by the Coulomb interaction.

In the 3D case, we consider a good metallic system with the localized d orbital, $v(\mathbf{q}) \sim \frac{1}{q^2}$, the electrons in the d orbital will be fully screened by surrounding charges.

Such an intrinsic screening effect will low down the total energy of the metallic system. As a result, the local on-site Coulomb interaction U will be decreased by the 3D fully screening when we compare this situation with the other (low dimensional) systems with inefficient screening.

But the situation is completely different for the 2D metal. If we set the 2D plane as the xy plane, $v(\mathbf{q}) \sim \frac{1}{q}$ and q is in the xy plane. The intrinsic screening is much weaker than that in the 3D case because the electrons can have a q_z dependent coupling, the Coulomb interaction is not fully screened in the direction of out-of-plane. As the Referee #2 mentioned, the substrate will also influence the screening effect, but this effect is extrinsic from the environments that we are not discussed in our manuscript. In such a case, compared with the 3D metallic state, the effective on-site Coulomb interaction U will become larger in the 2D limit.

On the other hand, the on-site Coulomb interaction U also depends on the kinetic energy of electrons around the Fermi level, which corresponds to the bandwidth of states on the Fermi level in 2D and 3D metal. The smaller the bandwidth is, the larger the on-site U is. The quantum confinement effect will decrease the bandwidth of metallic state in 2D, correspondingly to increase the on-site U . And such effect has been observed in our DFT calculations.

Moreover, in figure 3e and f, the authors calculate the MAE as a function of on-site Coulomb potential with the range of 0-8 eV to interpret the emergence of PMA (which needs to be further confirmed). However, I believe the screening of the substrate can hardly have a significant effect on ~7 nm metallic CrTe₂ due to such high carrier concentration.

In this part of our manuscript, we did not claim any effect on the substrate. We were talking about the change of on-site U induced by the dimensional effect, which is intrinsic due to the quantum physics of low-dimensional materials. We agree with Referee #2, the substrate cannot have a significant effect on these metallic thin film. But the quantum confinement in the 2D system will strongly modify the electronic structures around the Fermi level and is possible to tune the on-site Coulomb interaction U efficiently.

From the view of theoretical calculations, it is not easy to calculate the on-site Hubbard U precisely and quantitatively. More advanced many-body numerical calculation methods are required, which should be beyond mean-field approximation, such as the dynamical mean-field theory with constrained random phase approximation and density matrix embedding theory. In these calculations, the dimensional dependent screening effects should be considered seriously and the intrinsic on-site Coulomb interaction U should be calculated self-consistently. But such advanced calculations are beyond our discussion in this work. We change the on-site U artificially even in a large range and calculate the change of band structures and MAE. We did not expect that the experimental observations could cover the whole range, but the tendency of the change could provide some insight to understand the experimental observations.

Reviewer #3 (Remarks to the Author):

This work presents a comprehensive report on synthesis, structural characterisation and transport measurements on CVD grown CrTe₂. The authors claim observation of intrinsic ferromagnetism in the sample confirmed via robust Hall resistance measurement and demonstrating anomalous Hall effect. Layer dependent measurements show a transition in the magnetic easy axis from in-plane to out-of-plane. However, the experimental evidence for the above claims are not as convincing as one would like it to be.

We thank Reviewer #3 for his/her review and for providing the advice. We provided a detailed point-to-point response to address the following concerns.

Concerns:

1. Structural characterization using STEM-HAADF has been shown. However, the explanation on the analysis can be improved to help the readers understand the same.

We greatly appreciate Reviewer #3's careful review and helpful advice. As advised by the reviewer, we modified this part and added some discussions in the revised manuscript, which are highlighted in blue. We changed "*The atomic-resolution STEM-HAADF image (Fig. 2b) clearly shows that each Cr atomic column is surrounded by six Te atom columns arranging into a hexagonal lattice, consistent with the in-plane atomic configuration of the 1T phase. The corresponding fast Fourier transformation (FFT) pattern of Fig. 2b can be indexed as the [001] zone axis of 1T-CrTe₂. The cross-sectional STEM-HAADF imaging and EELS mapping (Figs. 2d - 2f) reveal that each monolayer slab consists of one layer of Cr atoms sandwiched between two layers of Te atoms, arranging into a characteristic Z-shaped construction, which matches well with the nature of octahedral coordination in the 1T phase. The stacking order of 1T-CrTe₂ is AA stacking. No obvious intercalated atoms were detected between two individual layers. The atomic-scale STEM analysis proves that the as-grown crystals are 1T-CrTe₂.*" to "*The atomic-resolution STEM-HAADF image (Fig. 2b) clearly shows that each Cr atomic column is surrounded by six Te atom columns arranging into a hexagonal lattice, consistent with the in-plane atomic configuration of the 1T phase. Te atom columns with higher atomic number (Z) present brighter contrasts than Cr columns with a lower atomic number in Z-contrast STEM-HAADF imaging, indicating the location of Te and Cr in this sample. The corresponding fast Fourier transformation (FFT) pattern of Fig. 2b can be indexed as the [001] zone axis of 1T-CrTe₂. The cross-sectional STEM-HAADF imaging and EELS mapping (Figs. 2d - 2f and Supplementary Fig. 8d - g) reveal that each monolayer slab consists of one layer of Cr atoms sandwiched between two layers of Te atoms, arranging into a characteristic Z-shaped construction, which matches well with the nature of octahedral coordination in the 1T phase. The schematic of 1T-CrTe₂ crystal structure is shown in the inserts of Fig. 2b and d with Cr in maroon and Te in yellow, respectively. The atomic-scale STEM analysis proves that the as-grown crystals are*

1T-CrTe₂ with the AA stacking order. No obvious intercalated atoms were detected between two individual layers. From the in-plane STEM Z-contrast image and cross-sectional ADF analysis, we concluded that the as-synthesized 1T-CrTe₂ belongs to the space group P $\bar{3}$ m1 and the lattice parameters are $a = b = 3.77 \text{ \AA}$, $c = 6.01 \text{ \AA}$.”

2. XPS data shows the oxidation state of the sample however, this does not conclude on the phase of the material. Raman analysis shows 2 peaks. A mere comparison with VTe₂ is not sufficient to conclude that this is 1T phase. It might be useful to do polarization dependence and also have an idea on the number of modes to be expected in XX and XY polarization modes to verify the phase of the sample. It could be 1T, 1T' or 2H.

We thank Reviewer #3 for his/her comment and helpful advice. We agree with Reviewer #3's opinion that the XPS data shows the oxidation state of the sample and the polarization dependence measurement is powerful methods to verify the phase structure. Alternatively, we have presented atomic-resolution STEM characterization, which is another powerful tool to identify the phase structure due to the nature of the Z-contrast STEM-HAADF technique and widely used in 2D systems [Adv. Mater. 30, 1803477 (2018)], to claim the 1T nature of obtained CrTe₂ crystals in our manuscript. The atomic-resolution STEM image of CrTe₂ shows the in-plane crystal structure of the hexagonal 1T phase with each hexagonally arranged Cr atom surrounded by six Te atoms (Figure 2b). In contrast, for the 2H phase, each Cr atom will be surrounded by three Te₂ columns for the in-plane atomic images. Even though the AA stacking of the 3R phase would cause similar projected imaging with the 1T phase along the c-axis, our cross-section STEM-HAADF imaging (Figure 2d) provides convincing evidence for the existence of 1T-CrTe₂ with the characteristic Z-shaped construction, which is quite different from 2H phase as well. The high symmetry of the in-plane atomic images can also preclude the 1T' phase. The contrast of two types of atoms presenting as light gray and white spots is because of their small and large atomic numbers (Z). Correspondingly, the cross-sectional STEM-HAADF imaging (Figs. 2d) reveal that each monolayer CrTe₂ consists of one layer of Cr atoms sandwiched between two layers of Te atoms, arranging into characteristic Z-shaped construction, further confirming the 1T phase of CrTe₂. To make our results more convincing, atomic-resolution EELS mapping (in Figure 2e) also has been performed and confirmed the pure Cr layers in the sandwich-like structure of 1T-CrTe₂. The simulated crystal structure is shown in the inset of Figure 2b with Cr in maroon and Te in yellow, respectively. Combining the images from in-plane and cross-section, the crystal structure of CrTe₂ matches well with the nature of octahedral coordination in the 1T phase and belongs to the hexagonal space group P $\bar{3}$ m1. This is strong evidence of the crystal structure. In the revised manuscript, we modified the descriptions and highlighted by blue.

3. The VSM measurements in Fig. 3 (a) and (b) show clear MH loops with hysteresis showing magnetism. A similar measurement has been reported (arxiv:1909.09797),

however no mention of this is made.

As advised by Reviewer #3, we added a brief introduction of this referential work as “*Similarly, the characteristic ferromagnetic M-H loops at 10 K of CrTe₂ flakes with various thicknesses under in-plane magnetic field are observed. Meanwhile, Faraday measurement indicates that the T_c in few-layered CrTe₂ is around 305 K, which is similar to the results from our SQUID measurement.*” in the VSM/SQUID measurement part and cited the reference correspondingly [Preprint at <https://arxiv.org/abs/1909.09797> (2019)].

4. In Fig. 3(b), one can observe an additional minor step-like feature at low field (nearly 1000 Oe or so), what is the reason for this feature? It would be good to explain this.

We thank Reviewer #3’s careful review and these issues are constructive. Generally, the additional minor step-like feature at a low field is usually observed in complicated magnetic systems with composite ordering, such as ferrimagnetic or ferromagnetic materials inclusion with different switching fields [Nature 542, 75-79 (2017); ACS Appl. Nano Mater. 2, 6809-6817 (2019); Nanoscale 10, 11028-11033 (2018); Sci. Rep. 9, 10793 (2019)]. Meanwhile, the existence of nanosized grains with large variations in composition, chemical order, stacking faults, the exchange coupling at the interfaces, and the magnetic domains/domain wall could further induce a step-like feature in the M-H curve [Nanoscale 10, 11028-11033 (2018); Adv. Funct. Mater. 30, 1910036 (2020)]. Noting that no obvious defects and stacking faults are observed from the STEM results and no step-like feature is founded in the thin samples. Thus, in this work, we infer that the additional minor step-like feature at a low field may come from a complex magnetic domain structure, which is usually observed in the relatively thick 2D magnetic materials [Adv. Funct. Mater. 30, 1910036 (2020); Nature Materials 17, 778–782 (2018)]. However, the details of the formation mechanism of the kink are still unrevealed, which is worthy of further study. We added the possible explanation to the step-like feature in the revised manuscript.

5. The data in Fig. 3(c) showing magnetism has no saturation. It is necessary to measure at lower temperature or at higher fields.

We thank Reviewer #3 for his/her question. As advised by Reviewer #3, we retest a 3 nm thick sample by SQUID and the result is shown in Figure R16. Noting that the SQUID result is considered to have a higher resolution and a better signal-to-noise ratio than VSM [Mater. Today 27, 107 (2019)]. As shown in Figure R16, clear and saturated magnetic loops are exhibited when the temperature is below T_c. We updated Fig. 3(c) by Figure R16 in the revised manuscript.

Figure R16. Magnetic hysteresis loops for 1T-CrTe₂ flakes with a thickness of ~ 3.0 nm under the magnetic field parallel to the c-axis of the crystal.

6. Surely there is a change in the easy axis as the number of layers is decreased however, to claim a trend it is important to have more than just 2-3 data points.

We thank Reviewer #3 for his/her constructive advice. We further performed the *M-H* characterization in the samples of about 5 nm and 35 nm thick to describe the trend on the change in the easy axis. Meanwhile, in order to eliminate the influence of the sample content, we normalized the remanence signal by the sample area. As shown in figure R17, the statistical coercive force and remanence as a function of sample thickness were presented. The statistical coercive force and remanence increase as the thickness increase. Especially, for the sample thickness thicker than about 10 nm, the extracted coercive force and remanence under in-plane magnetic field is larger than that under out-of-plane magnetic field. Relatively, for the sample thickness thinner than about 10 nm, the extracted coercive force and remanence under out-of-plane magnetic field is larger than that under in-plane magnetic field. Thus, the critical thickness is about 10 nm. We updated the data in the revised manuscript.

Figure R17. Remanence and coercive force at zero field for 1T-CrTe₂ samples of various thicknesses.

7. The Raman spectrum as a function of number of days has been shown; however, what is the thickness of the sample used for this? It is possible that the environmental degradation is slow for thick samples but for samples at very thin limits the degradation is quick. In such a case this can lead to spurious signals in the MH measurements and be

misleading.

We thank Reviewer #3 for this helpful comment. We agree with the reviewer that the thickness of samples is a very important factor when arguing about environmental stability. In the previous version, the thickness of the sample for environmental stability investigations was about 5 nm. To give a clear illustration, we retested a sample with a thickness of about 4.3 nm, as shown in Figure R18 (Supplementary Figure 7 in the revised manuscript). Figure R18 shows the OM images and corresponding Raman signals of 1T-CrTe₂ samples exposed to the atmosphere for 0 day, 5 days, 7 days, 9 days, 10 days, 12 days, 14 days, and 15 days, respectively. Meanwhile, to give more evidence that the sample has magnetic properties or not, we also do the RMCD test under different air-exposure times, as shown in Figure R18(e). After being exposed to the air for 5 days, the optical contrast and morphology of the sample did not change, and the Raman intensity and RMCD signal were comparable to those of the fresh one. When the exposure time is extended to more than 10 days, the optical contrast and morphology of 1T-CrTe₂ become lighter and rugged. The corresponding Raman signal intensities decrease and eventually vanish. Finally, after 15 days of exposure to the air, no RMCD and Raman signals are detected, indicating that the sample has degraded.

Ambient temperature and humidity conditions may have a significant impact on the stability investigations, thus we have also recorded the data simultaneously (Figure R18(d)) and added these data in the revised manuscript (Supplementary Figure 7).

We agree with the reviewer that the environmental degradation is slow for thick samples but for samples at very thin limits the degradation is quick. However, for the thin samples, the *M-H* measurements were coated by PMMA, thus the results are robust. In the revised manuscript, we added these evidences as Supplementary Figure 7 in the SI.

Figure R18. Environmental stability investigations. (a) Optical images of 1T-CrTe₂ samples exposed in the atmosphere for 0 day, 5 days, 7 days, 9 days, 10 days, 12 days, 14 days, and 15 days, respectively. (b) AFM image of the sample being tested. (c) The corresponding Raman spectrum from a. (d) Humidity and temperature data during the test. (e) RMCD signal as a function of exposure time under ambient condition.

8. The reason for increase in T_c with decrease in number of layers is not convincing. A more detailed explanation is required to understand this.

We thank Reviewer #3 to raise this issue. In order to provide a clear physical picture, let us review the theory about magnetism in the 2D system briefly. According to the Mermin-Wagner theorem [Phys. Rev. Lett. 17, 1133 (1966)], 2D isotropic ferromagnet processes the long-range magnetic order in the ground state only. At the finite temperature ($T > 0$), the long-range FM order will be destroyed by the thermal fluctuations. The anisotropic magnetic energy (such as MAE) will stabilize such long-range magnetic order at the finite temperature in the 2D limit, and Currie temperature T_c strongly depends on the value of MAE. The larger the MAE is in 2D ferromagnets, the larger the T_c is [Nature 456, 267 (2017)].

While, according to second-order perturbation theory, the MAE is determined by the coupling of d orbitals around the Fermi level induced by intrinsic spin-orbit coupling (SOC), whose strength is determined by the elements [Phys. Rev. B 92, 014423 (2015), Phys. Rev. 52, 1178 (1937)]. While the distribution of d orbitals around the Fermi level can be influenced by on-site Coulomb interaction U and the quantum confinement effect.

The on-site Coulomb interaction U depends on the dimension of the system [Quantum Theory of the Optical and Electronic Properties of Semiconductor, 5 Ed, H. Haug, and S. W. Koch, World Scientific; Many-Particle Physics, 3 Ed, G. D. Mahan, Springer]. Via changing the thickness of the sample from hundreds to tens of nanometers, we expect the dimensional crossover from 3D to 2D. On the other hand, with a decrease in the number of layers, the quantum confinement effect will be important, which could induce a great change of the distribution of d orbitals around the Fermi level and effective intrinsic on-site U . We argue that such kind of change will enhance the MAE with a decrease in the number of layers, and enhance T_c . While, when we further decrease the thickness of the sample to the 2D limit (atomic monolayer), the thermal fluctuation will be more important, which will decrease the T_c again.

To describe such effect precisely and quantitatively, more advanced many-body numerical calculations are required, which is beyond the mean-field approximation, such as the dynamical mean-field theory with constrained random phase approximation and density matrix embedding theory. In these calculations, the dimensional dependent screening effects should be considered seriously and the intrinsic on-site Coulomb interaction U should be calculated self-consistently. But such advanced calculations are beyond our discussion in this work.

9. To show anomalous Hall effect magnetotransport and RMCD have been used. However, it is not clear why RMCD is necessary. This does not seem to give any additional information. It can be moved to supplementary.

We thank Reviewer #3 to raise this issue. Yes, we have nearly the same conclusions from magnetotransport and RMCD measurements. The necessity of RMCD is originated from the following reasons. Firstly, RMCD has been proven to be a nondestructive and effective technique for probing the 2D magnetism [Nature 563, 94–99 (2018); Nature Materials 17, 778–782 (2018)]. Following this way, RMCD is useful for investigating the magnetic properties of metastable materials. Meanwhile, RMCD measurements can be done without any extra nanofabrication, thus can eliminate the doping from various sources, such as glue, solution, metal particles, which are unlikely to investigate the magnetic properties [Nature 563, 47–52 (2018); Preprint at <https://arxiv.org/abs/1912.06364> (2019)]. Secondly, the physical mechanism of the two methods is different [Nature 563, 47–52 (2018); Adv. Mater. 31, 1900065 (2019)]. The energy splitting of the spin-up and spin-down states that are associated with the moment is proportional to the strength and sign of the magnetic exchange as well as the magnetization. Circularly polarized $\hat{\sigma}^\pm$ light will cause a transition from a particular spin state when the spin-orbit coupling is present. Thus, when linearly polarized light is normally incident on and reflects off the magnetized material, the phase difference between the right-circularly polarized (RCP) and left-circularly polarized (LCP) light leads to a rotation of the linear polarization and induces ellipticity through reflectance magneto circular dichroism (RMCD). For magnetotransport measurement, it detects the electrical signal based on the Hall device.

Overall, it is necessary to do the RMCD test. Actually, in other words, the same claims obtained from magnetotransport and RMCD measurement can further confirm our conclusions. We highlighted the necessity of RMCD in the revised manuscript to help the readers understand the same.

10. At what temperature is the DFT calculation for MAE accounted for? Can you show the MAE at which the anisotropy is observed significantly?

We thank Reviewer #3 for his/her comments. The DFT calculation is an *ab initio* calculation for the ground state. In principle, all the magnetic quantity from DFT calculation is for the ground state with zero temperature at the mean-field level. On the other hand, the MAE is obtained from the force theorem [J. Magn. Magn. Mater. 159, 337 (1996)], which is defined as the total energy difference between different spin configurations (such as the magnetic moments along z direction and in the xy plane) at the ground state with zero temperature. This method is a standard way to estimate MAE value in the field of DFT calculations [Rev. Mod. Phys. 89, 025008 (2017)] and has been widely used in the studies of magnetic materials [Phys. Rev. Lett. 102, 257203 (2009), Nature 456, 267 (2017)].

We could use the MAE energy from the DFT calculation to estimate the critical temperature of the magnetic phase transition by using different models. In the mean-field level, Cheng Gong *et. al.*, use the 2D Heisenberg model with anisotropic MAE energy from DFT calculation to estimate T_c at the mean-field level [Nature 456, 267 (2017)]. They found that the MAE energy will strongly enhance T_c in the 2D limit.

11. The authors claim applications based on ferromagnetism - while the mobility is too low, this is highly unlikely. It is not clear that this is more advantageous over CVT or other methods. A comparison has to be clearly established with CVT and MBE grown samples to make such a claim.

We thank Reviewer #3 for his/her advice. We agree with the reviewer that such a claim is inappropriate especially with no comparison with CVT and MBE grown samples. We revised the manuscript to not touch such a claim and not mention too much in this paper. Even though the mobility is low, the value of the carrier mobility in 1T-CrTe₂ is in the same order of magnitude as some 2D metallic materials, as shown in Table R1. Thus, it may have some application potential and need to be further investigated.

Despite the mobility, the method of CVD has its advantages compared to CVT and MBE. For example, CVD is more suitable for large areas and high throughput sample preparation compared with CVT and MBE. Meanwhile, no epitaxial substrate and high cost are needed when preparing the samples. Importantly, the layer-controlled synthesis of 1T-CrTe₂ by CVD put forward not only in enriching the types of magnetic materials, but also providing more rooms for the integration of functional materials

(heterostructures).

Table R1. Comparison of the carrier mobility of 1T-CrTe₂ and other 2D materials.

Materials	Methods	Carrier mobility	Reference
1T'-MoTe ₂	Laser heating	$\sim 50 \text{ cm}^2\text{V}^{-1}\text{s}^{-1}$	Science, 349, 625-628 (2015)
Fe ₃ GeTe ₂	MBE	$\sim 54.9 \text{ cm}^2\text{V}^{-1}\text{s}^{-1}$	npj 2D Mater. Appl., 1, 1-7 (2017)
1T-TaS ₂	Micromechanical exfoliation	$\sim 20\text{-}15 \text{ cm}^2\text{V}^{-1}\text{s}^{-1}$	Nano Lett., 15, 1861–1866 (2015)
1T-CrTe ₂	CVD	$\sim 9 \text{ cm}^2\text{V}^{-1}\text{s}^{-1}$	This work

12. With the above concerns and prior work in this field, this work is more suitable for a journal like Scientific Reports or Phys. Rev. B after incorporating the corrections as suggested. This is certainly an interesting incremental work presenting details of synthesis of another 2D material to be added to the shelf.

We thank Reviewer #3 for his/her comments. As we know from prior work, the discovery of FM van der Waals materials has opened up unprecedented opportunities to explore new physical paradigms and develop novel spintronic devices. Up till now, the developed strategies to build 2D layered FMs or corresponding heterostructures are mostly based on methods with high fabrication cost and low throughputs, such as molecular beam epitaxy and mechanical exfoliation. Explicit synthesis of such materials using a facile and scalable strategy like CVD remains a challenge and the mechanisms of controlled synthesis of magnetic layered materials based on CVD are unclear.

Even though a large portion of the 2D society is trying to grow 2D ferromagnets by CVD, however, the magnetic properties of the obtained materials are uncertain and the AHE has rarely been observed. In this work, we demonstrate a CVD method to realize the thickness-controlled growth of 2D metallic 1T-CrTe₂ with long-range FM order. A robust AHE is observed in CVD synthesized 2D materials. We also observed some novel phenomena in this system: with the thickness of 1T-CrTe₂ reduced from tens of nanometers to several nanometers, the easy-axis changes from the direction of in-plane to out-of-plane; both magneto-transport and RMCD measurements verify the monotonic increase of T_c with thinner thickness, which is opposite to that of recently reported FM materials, such as CrI₃ and Fe₃GeTe₂. Our theoretical calculations indicate that the weakening of the Coulomb screening in the 2D limit may play a crucial role in the observed thickness-dependent magnetic properties of 1T-CrTe₂, providing a new strategy to tune the magnetic properties of ultrathin 2D materials.

In addition, as-synthesized 1T-CrTe₂ is relatively stable at ambient conditions and is processable in the air without any encapsulation, indicating its application potential in future spintronic devices. Thus, the simplicity and scalability of this CVD method provide an efficient strategy to construct 2D materials with exciting magnetic

properties and to build up 2D magnetic heterostructures, as well. We believe our work will be an inspiring contribution to the research in 2D materials, especially in 2D magnets.

Reviewers' comments:

Reviewer #1 (Remarks to the Author):

I would like to thank the authors for their careful and detailed responses to my queries, in particular for the presented statistical distributions of flakes and greater clarity in many areas.

Reviewer #2 (Remarks to the Author):

I appreciate the authors' efforts in performing additional experiments and discussion in response to my comments. However, the origin of the change in magnetic properties in thin flakes is still not clear, and I believe the degradation, which is indeed observed by the authors, plays an important role. This is supported by the following fact: In a recent work (Sun and et. al. arXiv:1909.09797, 2019), the exfoliated few-layer 1T CrTe₂ are encapsulated using h-BN in the glove box, which has been proved to be an efficient way to avoid air-degradation of ultra-thin air-sensitive flakes. The better protection of their sample is also supported by the observation of high T_c of ~316 K. However, they observed very different magnetic behavior from the author's observations here: In 10 nm CrTe₂, their sample shows strong in-plane anisotropy with almost 0 remanent magnetization along z, while the authors here observed strong out-of-plane anisotropy with quite large remanent magnetization along z (figure 3e) in the sample with the same thickness. Since Sun's sample is proved to be better protected, I tend to believe their observation is more reliable and intrinsic, which means the very different observation by the authors here is likely extrinsic and could be caused by the degradation. Therefore, I do not think it's convincing at all to just attribute the change in magnetic properties to the weakening of the Coulomb screening in the 2D limit. On the other hand, T_c above room temperature has been observed in both bulk and few-layer (~10 nm) 1T CrTe₂ (J. Phys.: Condens. Matter 27,176002, 2015; arXiv:1909.09797, 2019), while the T_c obtained here is well below RT for all thicknesses (< 213 K). The much worse sample quality and low T_c not only makes it dangerous to claim the delicate physical mechanism for the magnetic order but also defeats the point of application. In summary, I believe this work is neither solid in physics, nor useful in application. Therefore, I do not recommend the manuscript to be published in Nature Communications.

Reviewer #3 (Remarks to the Author):

I appreciate the efforts of the authors for giving a detailed response to all the concerns raised. Without any further objections on the scientific content, I recommend this work for publication in Nature Communications.

We thank the referees for the comments and have revised the paper accordingly to address the points raised. In this point-to-point response letter, comments from the referees are in black typeface, and our responses are in the blue typeface.

Reviewers' comments:

Reviewer #1 (Remarks to the Author):

I would like to thank the authors for their careful and detailed responses to my queries, in particular for the presented statistical distributions of flakes and greater clarity in many areas.

We thank Reviewer #1 for the positive evaluation and recommendation of the publication of this work.

Reviewer #2 (Remarks to the Author):

I appreciate the authors' efforts in performing additional experiments and discussion in response to my comments.

We greatly appreciate Reviewer #2 for the careful review and the constructive comments.

However, the origin of the change in magnetic properties in thin flakes is still not clear, and I believe the degradation, which is indeed observed by the authors, plays an important role. This is supported by the following fact: In a recent work (Sun and et. al. arXiv:1909.09797, 2019), the exfoliated few-layer 1T CrTe₂ are encapsulated using h-BN in the glove box, which has been proved to be an efficient way to avoid air-degradation of ultra-thin air-sensitive flakes. The better protection of their sample is also supported by the observation of high T_c of ~316 K. However, they observed very different magnetic behavior from the author's observations here: In 10 nm CrTe₂, their sample shows strong in-plane anisotropy with almost 0 remanent magnetization along z, while the authors here observed strong out-of-plane anisotropy with quite large remanent magnetization along z (figure 3e) in the sample with the same thickness. Since Sun's sample is proved to be better protected, I tend to believe their observation is more reliable and intrinsic, which means the very different observation by the authors here is likely extrinsic and could be caused by the degradation. Therefore, I do not think it's convincing at all to just attribute the change in magnetic properties to the weakening of the Coulomb screening in the 2D limit. On the other hand, T_c above room temperature has been observed in both bulk and few-layer (~10 nm) 1T CrTe₂ (J. Phys.: Condens. Matter 27,176002, 2015; arXiv:1909.09797, 2019), while the T_c obtained here is well below RT for all thicknesses (< 213 K). The much

worse sample quality and low T_c not only makes it dangerous to claim the delicate physical mechanism for the magnetic order but also defeats the point of application. In summary, I believe this work is neither solid in physics, nor useful in application. Therefore, I do not recommend the manuscript to be published in Nature Communications.

We thank Referee #2 for his/her careful comment. The biggest concern of the reviewer is that our experimental results are different from the one from reference [(Sun and et. al. arXiv:1909.09797, 2019)], and that our results might be extrinsic and could be caused by the degradation.

First of all, we would like to emphasize the difference in the sample synthesis methods. In the reference mentioned, samples are produced from KCrTe_2 by reaction with iodine in a solution environment. In this process, the fabricated CrTe_2 thin flakes might be heavily doped by electrons due to incomplete reaction. This is very similar to the exfoliation of LiMoS_2 in water, which will cause electron doping in MoS_2 and then induced a phase transition from H phase to T phase [Nat. Nanotechnol. 10, 313–318 (2015)]. The electron doping in two dimensional (2D) metallic ferromagnetic thin flakes, e.g. 1T- CrTe_2 , will definitely change its intrinsic magnetic properties, which has been widely seen in other 2D magnetic materials, like Fe_3GeTe_2 [Nature 563, 94–99 (2018)]. Meanwhile, in the mentioned reference, the sample is covered by Pt during the magneto-transport measurements, which might further have a great impact on the test results due the interfacial effect of the heavy metal [Nat. Nanotechnol. 14, 408–419 (2019)]. In sharply contrast, our samples are synthesized by chemical vapor deposition (CVD), in which CrTe_2 without apparent doping can be obtained, which can be proved by atomic-resolution STEM-HAADF results (Fig. 2). The cross-sectional STEM-HAADF imaging and EELS mapping clearly reveal that the as-grown crystals are CrTe_2 with 1T phase structure and no obvious intercalated atoms were detected between two individual layers. Thus, our observations are likely to be more intrinsic and reliable.

We agree with Reviewer #2 that using h-BN encapsulation in a glove box is an effective way to protect the ultra-thin air-sensitive flakes to avoid air-degradation. In order to further remove the concerns of the reviewer, we investigated the magnetic properties of 1T- CrTe_2 samples with and without h-BN encapsulation by RMCD, respectively. The results are shown in Figure R1. To prevent the sample from being exposed to the air, the samples are directly synthesized though the CVD equipment in the glove box. The h-BN encapsulated 1T- CrTe_2 samples were prepared in a glove box using the dry-transfer method [2D Mater. 1, 011002 (2014)]. The h-BN flakes were exfoliated on a PDMS substrate, which was used as a stamp. All the transfer processes are conducted on room temperature. The optical image of the encapsulated 1T- CrTe_2 flakes along with the un-encapsulated samples are shown in Figure R1 a, which are marked as CrTe_2 -BN and CrTe_2 , respectively. These flakes show uniform colors under the optical microscope, indicating uniform thicknesses.

Figure R1 b and c show the nearly vertical jumps of the RMCD signals versus perpendicular magnetic field below T_c , indicative of ferromagnetism of 1T-CrTe₂ flakes. For these two samples, the hysteresis loops shrink as the temperature increases and vanish at the magnetic transition point. The transition temperature from the ferromagnetic phase to the paramagnetic state can be estimated from the RMCD signal as a function of vertical magnetic field measured at several fixed temperatures. As shown in Figure R1 c, the extracted T_c of the samples with and without h-BN encapsulation are ~ 184 K and 185 K, respectively. Importantly, the T_c of the 1T-CrTe₂ sample protected by h-BN is almost the same as that of the un-encapsulated sample, which indicates that the T_c derived from the RMCD signals from the unprotected sample is reliable. Therefore, the change in T_c with the thickness cannot be attributed to sample degradation. Moreover, we observed strong out-of-plane anisotropy with quite a large remanent magnetization along the z-direction in the h-BN encapsulated sample, which is consistent with our previous observations. For the h-BN encapsulated sample, it is not exposed to the air before the measurements, so the intrinsic magnetic properties can be measured. The distinct difference between the samples prepared by the CVD method and the solution process indicates the material properties of different synthesis methods are also different.

Interestingly, the 1T-CrTe₂ flake covered by h-BN shows three-time larger coercive field than that in the samples without encapsulation. This might be caused by the interfacial effect between the 1T-CrTe₂ and h-BN, which deserves further studies. Again, the increase in T_c in thinner samples without h-BN covering cannot be attributed to the sample degradation, because thinner samples are more susceptible to oxidation in air [ACS Nano 14, 15256–15266 (2020) and Nat. Comm. 11, 3729 (2020)].

Figure R1. The investigation of magnetic performance of 1T-CrTe₂ samples with and without encapsulation by RMCD. (a) OM image of the samples being tested. Samples with and without encapsulation are highlighted by a blue and light blue hexagon, respectively. (b, c) RMCD signal as a function of the out-of-plane magnetic field at different temperatures obtained in 1T-CrTe₂ domains without and with encapsulation. (d) Corresponding zero-field remanent RMCD signal as a function of temperature for samples being tested. The solid lines are the least square fitting results of primary data using the formula: $\alpha(1-T/T_c)^\beta$. Zero RMCD signal is indicated by the dotted line.

In summary, the above experimental results and discussions show that our results are reliable and contributed by the intrinsic properties of this material. The difference between our results and those in the reference might be due to the difference in the sample synthesis method, where the CrTe₂ in reference [(Sun and et. al. arXiv:1909.09797, 2019)] might be heavily doped due to the incomplete reaction between KCrTe₂ and I₂. AHE, which is strongly dependent on the sample quality, is only measured in our paper. Also, the atomic-resolution STEM-HAADF image along with the EELS analysis (Fig. 2) clearly shows the atomic structure and the high crystal quality of the synthesized 1T-CrTe₂ single crystals.

Meanwhile, CVD is capable to grow large area and high-quality 2D materials, which has seldom been applied to 2D magnetic materials. Developing CVD to grow 2D magnetic materials will definitely be helpful for their practical applications.

The corresponding figures and discussions are added in the revised manuscript.

Reviewer #3 (Remarks to the Author):

I appreciate the efforts of the authors for giving a detailed response to all the concerns raised. Without any further objections on the scientific content, I recommend this work for publication in Nature Communications.

We thank Reviewer #3 for being satisfied with our revision and suggesting the publication of this work.

REVIEWERS' COMMENTS

Reviewer #2 (Remarks to the Author):

I appreciate the authors' efforts in performing additional experiments and discussion in response to my comments. Though I still have some concerns about the different magnetic properties in the CrTe₂ samples from different synthesis methods, which I think is still a question to be explored, the consistent magnetic behavior in CrTe₂ samples with and without BN encapsulation certainly makes the authors' conclusion more convincing.

I have no further comments and would like to recommend publication in Nature Communications.